# Minimal metabolic pathway structure is consistent with associated biomolecular interactions

Aarash Bordbar[1], Harish Nagarajan[2], Nathan E Lewis[1,3,4], Haythem Latif[1], Ali Ebrahim[1], Stephen Federowicz[2], Jan Schellenberger[2] & Bernhard O Palsson[1,2,5,*]

## Abstract

Pathways are a universal paradigm for functionally describing cellular processes. Even though advances in high-throughput data generation have transformed biology, the core of our biological understanding, and hence data interpretation, is still predicated on human-defined pathways. Here, we introduce an unbiased, pathway structure for genome-scale metabolic networks defined based on principles of parsimony that do not mimic canonical human-defined textbook pathways. Instead, these minimal pathways better describe multiple independent pathway-associated biomolecular interaction datasets suggesting a functional organization for metabolism based on parsimonious use of cellular components. We use the inherent predictive capability of these pathways to experimentally discover novel transcriptional regulatory interactions in *Escherichia coli* metabolism for three transcription factors, effectively doubling the known regulatory roles for Nac and MntR. This study suggests an underlying and fundamental principle in the evolutionary selection of pathway structures; namely, that pathways may be minimal, independent, and segregated.

**Keywords** constraint-based modeling; genetic interactions; pathway analysis; protein-protein interactions; transcriptional regulatory networks
**Subject Categories** Genome-Scale & Integrative Biology; Metabolism
**Mol Syst Biol.** (2014) 10: 737

## Introduction

Historically, biochemical experimentation has defined pathways or functional groupings of biomolecular interactions. Such pathways are foundational to human-curated databases, such as KEGG (Kanehisa *et al*, 2012), BioCyc (Caspi *et al*, 2010), and Gene Ontology (Ashburner *et al*, 2000), are the basis for education in biochemistry, and are broadly deployed for analyzing and conceptualizing complex biological datasets (Khatri *et al*, 2012). However, the order of discovery and perceived importance of cellular components has unavoidably introduced a man-made bias. Pathway organization is thus often defined in a universal (rather than organism-specific) manner, missing potential organism-specific physiology. It is unclear whether the currently used pathway structures correctly account for observed interactions between the macromolecules needed to carry out their function.

Systems biology has led to the elucidation and analysis of multiple cellular networks, representing metabolism (Mo *et al*, 2009; Orth *et al*, 2011), transcriptional regulation (Gama-Castro *et al*, 2011), protein-protein interactions (Han *et al*, 2004), and genetic interactions (Costanzo *et al*, 2010). These networks provide the opportunity to build unbiased pathway structures using statistical or mechanistic algorithms. Statistical approaches have been employed to high-throughput data and interaction networks to reconstruct the cellular component ontology of Gene Ontology (Dutkowski *et al*, 2012). However, such approaches were not meant to reconstruct the Biological Processes ontology and build pathways (Dolinski & Botstein, 2013).

Mechanistic approaches include utilizing convex analysis with metabolic networks to automatically define pathways. Genome-scale metabolic networks contain curated and systematized information about all known biochemical moieties (metabolites) and transformations (reactions) of a particular cell's metabolism encoded on its genome and described in experimental literature (Feist *et al*, 2009). The stoichiometric matrix (**S**) is a mathematical description of a genome-scale metabolic network, which can be queried by many available modeling methods (Lewis *et al*, 2012). These models and the calculated reaction fluxes are typically studied under a steady-state assumption (Fig 1A). Thus, the full set of potential steady-state reaction fluxes of a metabolic network is contained in the associated null space of **S** (Palsson, 2006). The basis vectors of the null space have been previously shown to correspond to biochemical pathways providing a fundamental connection between mathematical and biological concepts (Papin *et al*, 2003). This connection has generated many attempts to characterize the null

1 Department of Bioengineering, University of California San Diego, La Jolla, CA, USA
2 Bioinformatics and Systems Biology Program, University of California San Diego, La Jolla, CA, USA
3 Department of Pediatrics, University of California San Diego School of Medicine, La Jolla, CA, USA
4 Wyss Institute for Biologically Inspired Engineering and Department of Genetics, Harvard Medical School, Boston, MA, USA
5 Novo Nordisk Foundation Center for Biosustainability, Technical University of Denmark, Lyngby, Denmark
*Corresponding author. Tel: +1 858 534 5668; Fax: +1 858 822 3120; E-mail: palsson@ucsd.edu

space's contents using convex analysis (Clarke, 1980). Though readily applicable to small networks, it has been recognized for some time that convex pathway definitions (e.g., extreme pathways (Schilling *et al*, 2000) and elementary flux modes (Stelling *et al*, 2002)) cannot be globally applied to genome-scale networks as the enumeration of all such pathway vectors is not computationally feasible (Yeung *et al*, 2007). More recently, approaches have been developed to define subsets of metabolic pathways (de Figueiredo *et al*, 2009; Kaleta *et al*, 2009; Kelk *et al*, 2012), though these pathways do not describe the totality of phenotypic states.

In this study, we present a mixed-integer linear optimization algorithm (MinSpan) that can for the first time define the shortest, functional pathways for metabolism at the genome scale using metabolic networks thereby describing the totality of steady-state

phenotypes. We find that (1) the minimal pathways are biologically supported by independent biomolecular interaction networks, (2) the minimal pathways have stronger biological support than traditional human-defined metabolic pathways, and (3) the minimal pathways guided experimental discovery of novel regulatory roles for *E. coli* transcription factors.

## Results

### Defining a minimal network pathway structure for metabolism

In this study, we introduce a network-based pathway framework called MinSpan that calculates the set of shortest pathways (based on reaction number) that are linearly independent from each other

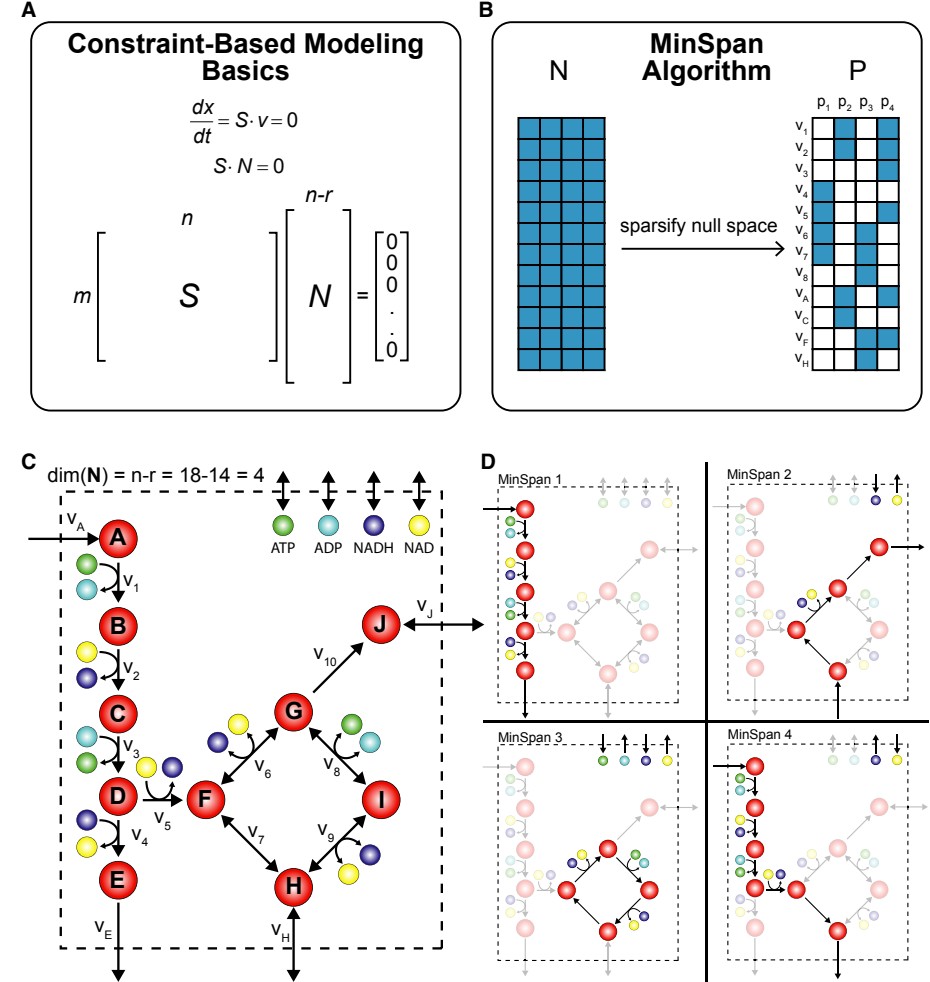

**Figure 1. Overview of the MinSpan algorithm.**

A  A metabolic network is mathematically represented as a stoichiometric matrix (**S**). Reactions fluxes (**v**) are determined assuming steady state. All potential flux states lie in the null space (**N**).

B  The MinSpan algorithm determines the shortest, independent pathways of the metabolic network by decomposing the null space of the stoichiometric matrix to form the sparsest basis.

C  A simplified model for glycolysis and TCA cycle is presented with 14 metabolites, 18 reactions, and a 4-dimensional null space. Reversible reactions are shown.

D  The four pathways calculated by MinSpan for the simplified model are presented, two of which recapitulate glycolysis and the TCA cycle, while the other two represent other possible metabolic pathways. The flux directions of a pathway through reversible reactions are shown as irreversible reactions.

(Fig 1). The MinSpan pathways are the sparsest linear basis of the null space of $S$ that maintains the biological and thermodynamic constraints of the network. The MinSpan pathways have a couple notable properties. First, unlike convex analysis approaches (Llaneras & Pico, 2010), MinSpan pathways can be computed for genome-scale metabolic networks. Second, the sparsest basis (Fig 1B) maximally segregates the network into clusters of reactions, genes, and proteins that function together. This property allows for an unbiased functional segregation of cellular metabolism into biologically meaningful pathways.

The mathematical derivation of MinSpan is provided in the Materials and Methods. Here, we begin with an illustrative example of MinSpan for a metabolic model that contains a simplified representation of glycolysis and the TCA cycle (Fig 1C). In this example, $S$ has dimensions ($m \times n$) where $m = 14$ metabolites and $n = 18$ reactions. The linear basis for the null space ($N$) has dimensions ($n \times n - r$) where $r$ is the rank of $S$. This $S$ has rank ($r = 14$), meaning that the null space is four dimensional (e.g., =18–14). Thus, a set of four linearly independent pathways through the network represents a linear basis for the null space of $S$. There are numerous potential sets of linearly independent pathways for a metabolic network as the linear basis of the null space is not unique. MinSpan chooses a set representing the shortest independent pathways, and we later show that this linear basis is more biologically relevant than other linear bases.

Running MinSpan on the simplified model converts the linear basis matrix ($N$) to a MinSpan pathway matrix ($P$) that contains the four shortest, linearly independent reaction pathways (Fig 1B). The resulting pathways are presented on the network map (Fig 1D). MinSpan pathways #1 and #3 are similar to traditional metabolic pathways (e.g., pathways that look like glycolysis and TCA cycle in this simplified network), while the last two MinSpan pathways do not mimic traditional pathways. In the Supplementary Information, we contrast MinSpan with past convex analysis methods (e.g., Extreme Pathways and Elementary Flux Modes) and also present another illustrative but more complex example for *E. coli* core metabolism.

## MinSpan pathways are supported by independent biological datasets

MinSpan pathways are a fundamental and unbiased attempt to define pathways for metabolism. We next determined whether MinSpan pathways have biological relevance. By definition, pathways represent a grouping of biochemical transformations that can concurrently function. The biomolecular machinery (e.g., genes and proteins) of metabolic pathways has been previously shown to preferentially share interactions compared to components outside of pathways. Thus, the genes within pathways preferentially contain positive genetic interactions (Kelley & Ideker, 2005) and are co-regulated (Wessely *et al*, 2011). Furthermore, the proteins within pathways preferentially contain protein-protein interactions (see Supplementary Information). Thus, we compared calculated MinSpan pathways of genome-scale metabolic networks to the independent genome-scale networks of protein-protein interactions (PPI) (Stark *et al*, 2006), genetic interactions (Costanzo *et al*, 2010), and transcriptional regulation (TRN) (Gama-Castro *et al*, 2011).

We computed MinSpan pathways for the genome-scale metabolic networks of *Escherichia coli* (Orth *et al*, 2011) and *Saccharomyces cerevisiae* (Mo *et al*, 2009). They contain 750 and 332 pathways, respectively, representing the dimensions of the two null spaces (see Supplementary Dataset S1). For each calculated MinSpan pathway, we grouped the "gene-protein-reaction" (GPR) associations (Fig 2A) of the metabolic reactions within that pathway (Fig 2B). The GPR association is a set of Boolean rules describing the required genes, transcripts, and proteins required to catalyze a metabolic reaction.

We hypothesized that a highly correlated co-occurrence or co-absence of two proteins across all the MinSpan protein sets of a particular organism was an indication that the proteins share a PPI and that a co-occurrence or co-absence of two genes implies that the genes positively interact or that they are co-regulated by the same transcription factor (TF). We compared MinSpan pathways to PPI and genetic interactions in *S. cerevisiae* and the TRN of *E. coli* as the datasets are most complete for those particular organisms (Fig 2C).

By testing for significant Spearman correlation coefficients of co-occurrence or co-absence of two proteins across the *S. cerevisiae* MinSpan pathway protein sets, 80% of known yeast two-hybrid PPIs in metabolism (Stark *et al*, 2006) were found within MinSpan pathways (Fig 2D). Similarly, MinSpan pathways were representative of positive genetic interactions (Costanzo *et al*, 2010) in metabolism based on significant correlations of genes across the *S. cereivisae* MinSpan pathway gene sets (Fig 2E).

We also used correlation analysis with *E. coli* pathway gene sets to assess consistency with co-regulation by the same TF (Fig 2F). The pathways share over 6,700 co-regulated gene pairs within *E. coli* metabolism. Our analysis quantitatively revealed two levels of regulation. First, local regulation (by TFs regulating at most 30 metabolic genes, which accounts for 90% of *E. coli* TFs regulating metabolism) is pathway based with TFs acting directly on linearly independent, minimal pathways (Fig 2F). Second, global regulation (TFs with more than 30 regulated metabolic genes) involves many simultaneous cellular functions that are not just metabolic and does not necessarily mimic the metabolic scaffold. Hence, MinSpan pathways recapitulate local and intermediate regulatory mechanisms, but do not capture the less specific roles of global regulators.

## MinSpan pathways are more biologically supported than human-defined pathways

MinSpan pathways are highly consistent with PPI, positive genetic interactions, and local transcriptional regulation implying their biological relevance. However, a key question arises: Are there other pathway structures (human or network-defined) that are equally or better suited at representing pathway-associated biomolecular data types?

To answer this question, we compared the biological relevance of MinSpan pathways to other network pathway structures derived from the null space of the stoichiometric matrix. We calculated the "MaxSpan" or a null space basis matrix with the least number of non-zero entries (e.g., the longest pathways) and generated "RandSpan" or randomly generated null space bases ($n = 100$) that had random criteria for the sparsity of the matrix.

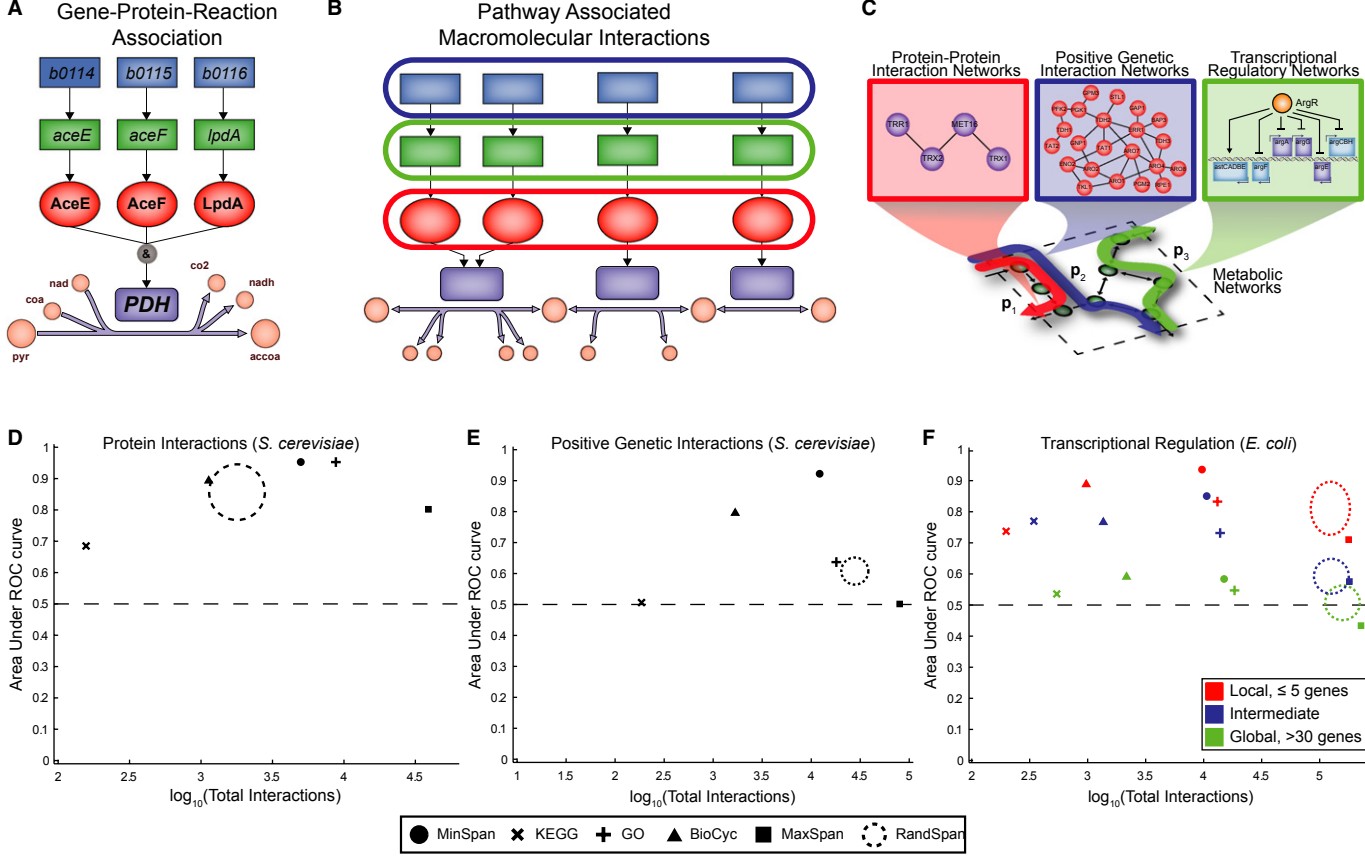

**Figure 2.  Correlation analysis shows MinSpan pathways are biologically relevant.**

A     "Gene-protein-reaction" (GPR) associations describe the necessary genes and proteins required for the catalysis of a metabolic reaction. Pyruvate dehydrogenase in *Escherichia coli* is shown as an example.

B     We grouped genes and proteins in the GPRs for each MinSpan pathway to check consistency with datasets on pathway-associated biomolecular interactions.

C–F   Correlation analysis (C) of the gene and protein sets shows MinSpan pathways are biologically consistent with three different biomolecular interaction networks: (D) protein-protein interactions in *S. cerevisiae* (yeast two-hybrid data), (E) positive genetic interactions in *S. cerevisiae* ($P < 0.05$ and $\varepsilon > 0.16$ as defined by Costanzo *et al*), and (F) transcriptional regulation in *E. coli*. MinSpan pathways are more consistent with data-driven protein interaction, genetic interaction, and transcriptional regulatory networks than human-defined pathways (KEGG, BioCyc, and GO), a least sparse null space (MaxSpan), and randomly generated null spaces (RandSpan). Accuracy (y-axis) is determined by the area under the curve (AUC) of the receiver operating characteristic (ROC) curve. Coverage (x-axis) is determined by the number of interactions the method made a prediction for. The dotted circle for RandSpan represents the mean plus one standard deviation of the 100 random null spaces. x- and y-axes values are in the Supplementary Information.

We also compared MinSpan to commonly used human-defined pathway databases including: KEGG Modules (Kanehisa *et al*, 2012), BioCyc (EcoCyc (Keseler *et al*, 2011) for *E. coli* and Yeast-Cyc (Cherry *et al*, 2012) for *S. cerevisiae*), and the Biological Processes ontology of Gene Ontology (Ashburner *et al*, 2000) for both organisms.

Surprisingly, repeating the correlation analysis for these alternative network and human-defined pathways, we found that MinSpan pathways were generally more consistent with recapitulating biomolecular interactions (Fig 2D, E, & F). For PPIs, MinSpan was marginally, but not statistically, more consistent than other methods ($P = 0.0780$ versus KEGG, $P = 0.168$ versus EcoCyc, $P = 0.901$ versus Gene Ontology, two-tailed *t*-test). As most PPIs occur between proteins within the same metabolic complex or adjacent metabolic reactions, most metabolic pathway structures should conserve PPIs. However, for positive genetic interactions ($P = 3.16\mathrm{e}{-3}$ versus KEGG, $P = 0.133$ versus YeastCyc, $P = 1.47\mathrm{e}{-3}$

versus Gene Ontology), local transcriptional regulation ($P = 2.79\mathrm{e}{-4}$ versus KEGG, $P = 0.181$ versus EcoCyc, $P = 2.35\mathrm{e}{-4}$ versus Gene Ontology), and intermediate transcriptional regulation ($P = 3.38\mathrm{e}{-3}$ versus KEGG, $P = 3.97\mathrm{e}{-6}$ versus EcoCyc, $P = 3.30\mathrm{e}{-18}$ versus Gene Ontology), MinSpan pathways were statistically more representative of the interactions. None of the pathway structures were highly consistent with global regulation.

These finding have two important implications. First, MinSpan pathways have more underlying support from biomolecular data types than human-defined pathways suggesting an alternative and fundamental modular organization of cellular metabolism. Defining pathways by human intuition and interpretation is less representative of the biomolecular interactions. Second, a minimal pathway structure is more biologically relevant than other potential linear bases of the null space confirming the principle underlying its use. The specific values in Fig 2 are available in the Supplementary Information.

### Global comparison of MinSpan and human-derived pathways

How different are the MinSpan pathways from other sources of pathway definitions? We can delineate the coverage and similarity of MinSpan pathways against traditional pathway databases (i.e., KEGG, BioCyc, and Gene Ontology) for *E. coli* and *S. cerevisiae* metabolism (Fig 3) to answer this question. First, we determined the number of pathways in each database covering the metabolic genes in the *E. coli* and *S. cerevisiae* metabolic models (see Table 1). There are widely varying numbers of pathways between all the databases. KEGG is the smaller of the two metabolic pathway databases. Gene Ontology contains many other genetic classifications and is larger than KEGG and BioCyc. MinSpan was the largest pathway database.

Second, we calculated pairwise connection specificity indices (CSI) (Green *et al*, 2011; Bass *et al*, 2013) for pathways across all databases (MinSpan and human-defined) based on their gene products and hierarchically clustered them (Fig 3A). The CSI provides both a metric of how similar two pathways are, and how specific their similarity is compared to the rest of the available pathways. For each pathway definition (MinSpan, KEGG, BioCyc, and GO), we determined how many of their pathways are captured in other pathway definitions by whether or not they shared a high CSI value (Fig 3B). As the number of MinSpan pathways is much larger than the number of pathways for other databases for *E. coli*, MinSpan captures most of their information, while KEGG, EcoCyc, and GO capture much less of MinSpan. For *S. cerevisiae*, Gene Ontology has the highest coverage. It has two fewer pathways than MinSpan (Table 1), but fully captures MinSpan pathways.

Third, we used a K-nearest neighbor search to assign the individual pathways of one pathway definition into the other three pathway definitions (Fig 3C) to determine whether certain pathway definitions are more similar than others. The number of pathways that were similar between KEGG and BioCyc pathways and BioCyc and GO was statistically significant ($P < 0.05$, binomial distribution, Bonferroni correction) for both organisms. The number of pathways that shared similarities with MinSpan was significantly depleted ($P < 0.05$, binomial distribution, Bonferroni correction), or dissimilar, from most pathway definitions in both organisms. This suggests that KEGG and BioCyc have the most similar pathways, followed with Gene Ontology, which has more similarities with BioCyc than KEGG. MinSpan pathways are significantly different from human-defined pathway databases and for *E. coli* contain many unique pathways (Fig 3D).

Fourth, we determined what caused the significant difference in MinSpan and human-defined databases by looking at the individual pathways that were similar or dissimilar between the pathway definitions (Fig 3D). For both *E. coli* and *S. cerevisiae*, MinSpan captured traditional pathways in carbon metabolism (e.g., glycolysis, pentose phosphate pathway, TCA cycle), amino acid metabolism, and nucleotide metabolism. However, 26 of the 56 pathways missed by MinSpan for *E. coli* were related to fatty acid metabolism. A MinSpan pathway operates under the steady-state assumption, leading to full flux balance of the metabolic network (e.g., all metabolites and cofactors in the pathway must be produced, consumed, and/or recycled). Traditional fatty acid pathway representations do not include all necessary components, and fatty acid pathways typically require the most precursors and cofactors. Conversely, 99 of the 204 MinSpan pathways missed by traditional pathways dealt with pathways that contained the necessary cofactors and precursors, mainly for fatty acid metabolism. The second representative difference was that a few traditional pathways (representing gluconeogenesis and deoxyribonucleotide biosynthesis) were broken up into smaller MinSpan pathways. Third, MinSpan pathways for *E. coli* contained 54 novel pathways related to ion transport, alternate carbon metabolism, and electron transfer.

For *S. cerevisiae*, Gene Ontology has a larger coverage of metabolism than MinSpan. This difference is due to two reasons: (1) the metabolic model of *S. cerevisiae* is not as comprehensive as the model for *E. coli* and (2) the Gene Ontology for *S. cerevisiae* is relatively more comprehensive than the one for *E. coli*. There were 32 pathways missed by MinSpan due to differing representations than traditional pathway databases. Seven missed pathways dealt with fatty acid metabolism, and their MinSpan counterparts took cofactors and precursors into account. Five traditional pathways for tyrosine biosynthesis and triglyceride biosynthesis were broken up into smaller pathways by the MinSpan algorithm. The specific pathways that are missed by the MinSpan algorithm are provided in the Supplementary Dataset S2.

### Key examples of MinSpan differences

From a global perspective, there are three representative differences between MinSpan and traditional pathways (Fig 4). First, MinSpan enumerates pathways not already described in databases. We found 54 metabolic pathways in *E. coli* that were not described in KEGG, Gene Ontology, or EcoCyc. For example, one such pathway involves the degradation of shikimate, an aromatic compound, to L-tryptophan (Fig 4A). The pathway consists of eight metabolic reactions, six of which are co-regulated by TrpR in *E. coli*, lending support to the pathway's biological relevance.

**Table 1.  Pathway numbers and lengths for MinSpan and pathway databases in *Escherichia coli* and *Saccharomyces cerevisiae*.**

| | *E. coli* | | | | *S. cerevisiae* | | | |
|---|---|---|---|---|---|---|---|---|
| | KEGG | EcoCyc | Gene Ontology | MinSpan (filtered) | KEGG | YeastCyc | Gene Ontology | MinSpan (filtered) |
| Number of pathways | 91 | 199 | 348 | 737 | 74 | 121 | 296 | 298 |
| Average pathway length (number of genes) | 5.4 | 6.6 | 7.3 | 13.6 | 3.9 | 7.1 | 7.9 | 13 |
| Average gene usage | 1.6 | 2.7 | 2.4 | 8.9 | 1.4 | 2.2 | 6.1 | 7.1 |

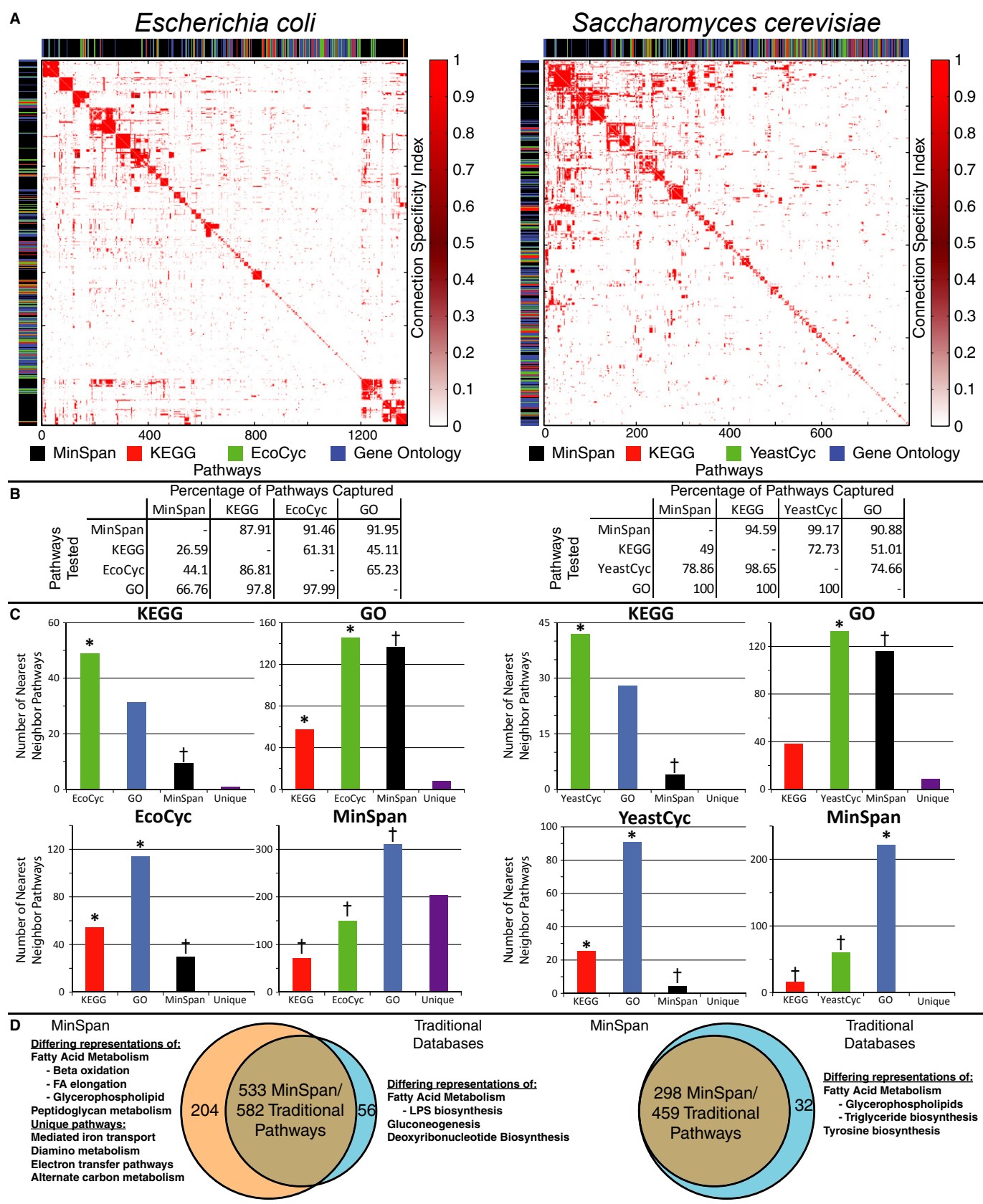

Figure 3.

◄

**Figure 3.   Global comparison of MinSpan pathways with databases of human-defined pathways.**

A   The pairwise connection specificity index (CSI) was calculated for all pathway definitions (from four sources: MinSpan, KEGG, BioCyc, and Gene Ontology) for *Escherichia coli* and *Saccharomyces cerevisiae* as a measure of pathway similarity. The CSI matrix was hierarchically clustered and the database that the pathway originates from is color-coded to the left and above the heatmap to illustrate the clustering.

B   The percentage of pathways that share a high CSI value (top 15% of interactions) between pathway databases is presented. For example, MinSpan pathways are similar to and capture roughly 88% of all pathways in KEGG for *E. coli*. Conversely, KEGG is much smaller and only captures 26% of MinSpan pathways.

C   A K-nearest neighbor search was done to see how pathways classify into other databases. KEGG and BioCyc pathways have the closest resemblance with Gene Ontology being the next similar. MinSpan is significantly different than human-defined pathways. (*Significant enrichment, †Significant depletion).

D   533 MinSpan pathways are similar to 582 traditional pathways. There is not a one-to-one mapping as similar pathways may exist in multiple human-defined databases. For *E. coli*, there are 204 unique MinSpan pathways. In *S. cerevisiae*, there are none as Gene Ontology captures all the pathways from the metabolic model.

Second, MinSpan pathways are mass-balanced and are functionally independent units that take into account systemic requirements. For example, the traditional human-defined pathway for purine biosynthesis starts from phosphoribosyl pyrophosphate (prpp) and L-glutamine. Purine biosynthesis consists of 11 metabolic reactions that lead to IMP (Fig 4B), which is further modified to other purines. This traditional pathway exists in KEGG, Gene Ontology, and EcoCyc and is biologically relevant as all 11 reactions are co-regulated by PurR in *E. coli*. The third metabolic reaction for purine biosynthesis (phosphoribosylglycinamide formyltransferase) requires 10-formyltetrahydrofolate (10fthf). Thus, the MinSpan pathway for purine biosynthesis also includes tetrahydrofolate (THF) recycling which contains three reactions. The gene for the first reaction in THF recycling is transcriptionally regulated by PurR, while the two other reactions' genes are not transcriptionally regulated. MinSpan elucidates a coupling of THF recycling to IMP biosynthesis that is independently verified by the co-regulation of the necessary genes.

Third, the minimalist decomposition of MinSpan is especially useful for complex metabolic network topologies where pathway enumeration is manually difficult. For example, threonine and methionine metabolism in *S. cerevisiae* is a small but complex network consisting of 12 metabolic reactions that involve multiple amino acids (Fig 4C). KEGG contains two pathways for this region: L-aspartate to L-threonine and L-aspartate to L-methionine (Fig 4C). This ignores another potential path to L-methionine from L-threonine. YeastCyc and GO cover all reactions in the example by containing many more pathways, seven and five pathways, respectively. Similar to KEGG, both YeastCyc and GO describe L-methionine synthesis through L-aspartate. On the other hand, MinSpan decomposes the network into L-threonine production through L-aspartate and L-methionine production through L-threonine (Fig 4C). These two functional units contain the shortest possible connection between the major metabolites. In the process, this decouples L-aspartate from L-methionine production.

Genetic interactions are consistent with the parsimonious approach. From the correlation analysis (at a False Positive Rate of 20%), both MinSpan and human-defined pathways correctly identified four positive genetic interactions in the L-threonine synthesis pathway (Fig 4C), suggesting a functional metabolic pathway. However, there were no positive genetic interactions in the traditional L-methionine synthesis from L-aspartate, suggesting no functional pathway and leading to five false positive predictions. In fact, YDR158W and YER091C interact negatively, further supporting that the two genes are not in the same pathway. Conversely, MinSpan separates L-aspartate from L-methionine and hence correctly predicts no genetic interactions. In addition, YCL064C negatively interacts with YER052C, YDR158W, YJR139C, and YCR053W lending support to L-threonine and L-methionine production being decoupled.

**MinSpan pathways predict transcriptional regulation**

MinSpan is an inherent property of metabolic networks, unlike human-defined pathways, and offers the direct ability to predict pathway-associated biomolecular properties from flux distributions calculated by constraint-based modeling. From the above correlation analysis, we observed that genes within a MinSpan pathway are often co-regulated by the same TF. Thus, we tested whether TFs act directly on the MinSpan pathway structure during metabolic shifts to coordinate expression of the metabolic genes needed to implement a fully functional pathway.

Constraint-based models can be used with Monte Carlo sampling methods to compute candidate reaction flux states through the metabolic network (Schellenberger *et al*, 2011). Comparing the significantly changed reaction fluxes between two sampled metabolic conditions has been previously shown to be consistent with experimental datasets (Lewis *et al*, 2010; Bordbar *et al*, 2012; Nam *et al*, 2012) (Fig 5A). However, these predicted differences are on an individual reaction basis, not for coordinated changes in flux states that might reflect the actions of the TRN.

A reaction flux state can be decomposed into its constituent pathways. As the MinSpan pathway matrix (**P**) is a linear basis for the null space of **S**, any sampled flux distribution (**v**) can be decomposed into linear weights ($\alpha$) of **P** (Wiback *et al*, 2003). Thus, metabolic reactions (**v**) determined from Monte Carlo sampling can be converted into changes in pathway flux loads ($\alpha$) (Fig 5A). As MinSpan pathways maintain the transcriptional regulatory hierarchy, the MinSpan pathways can then be associated to TFs based on enrichment of known regulatory gene targets (Gama-Castro *et al*, 2011) ($P < 0.01$, hypergeometric test). Thus, a significant change in the flux load of a MinSpan pathway ($\alpha$) is a direct predictor of pathway-associated TF activity.

Metabolic reaction fluxes were computed by Monte Carlo sampling (Schellenberger *et al*, 2011) for minimal, aerobic glucose conditions, as well as 51 nutritional shifts due to changes in carbon, nitrogen, phosphorus, and sulfur sources, as well as supplementation of amino acids and nucleotide precursors and removal of oxygen. Sampled metabolic reaction fluxes (**v**) were decomposed into MinSpan pathway flux loads ($\alpha$) to determine significantly changed pathways across nutritional conditions (Fig 5A). TFs associated with significantly changed pathways ($P < 0.05$, Wilcoxon signed-rank test) were then used as predictors of transcriptional regulation.

Predicted TF activities (Fig 5B) substantially agreed with known regulatory changes detailed in EcoCyc (Keseler *et al*, 2011) and primary literature (see Supplementary Information). As the TRN is not completely known, we focused primarily on true-positive and false-negative results. Transcriptional regulatory changes for 37 of

**Figure 4.   The three differences between MinSpan and human-defined pathways.**

A   MinSpan automates the enumeration of biologically relevant pathways.

B   MinSpan includes all required components of a pathway to be independent. The additional pathway components not found in human-defined pathways, such as THF recycling, are often co-regulated and thus a part of a coherent pathway functioning as a "module" in a network.

C   MinSpan decomposes complex topology into the simplest representation. For example, there is a shorter route to L-methionine production through L-threonine than from L-aspartate. Note: For MinSpan pathways, only the representative genes of the pathway are shown. 10fthf: 10-formyltetrahydrofolate; 2obut: 2-oxobutanoate; gar: glycinamide ribonucleotide; L-cysta: L-cystathionine; methf: 5,10-methenyltetrahydrofolate; mlthf: 5,10-methylenetetrahydrofolate; prpp: phosphoribosyl pyrophosphate; skm: shikimate; thf: tetrahydrofolate.

the 51 nutrient shifts matched known associations and eight other shifts partially matched known TF–environment associations (see Supplementary Dataset S3). Overall, the predicted activities were highly enriched in known TF–environment associations ($P = 2.79e-107$, binomial distribution). Focusing on the 45 shifts, there were 247 predicted TF–environment associations. 154 of those predictions are confirmed in EcoCyc and primary literature. 93, or 38%, of the predicted TF activities are not known to be associated with the corresponding shift, providing numerous novel transcriptional regulation predictions.

Hierarchically clustering nutrient shifts based on predicted TRN response stratifies key classes of shifts (Fig 5B). Nucleotide precursor supplementation is characterized by PurR and GcvA activity. Alternate sulfur sources as well as L-cysteine and L-methionine supplementation clustered by CysB activity. Sugar carbon sources clustered by Cra. Organic acid carbon sources, including glycerol, were systemic and characterized by Fnr, Lrp, and Cra activity. Other systemic shifts included the response to the lack of oxygen and

alternate nitrogen sources. Finally, predicted transcriptional regulatory changes of well-studied shifts (Cho *et al*, 2011, 2012) of amino acid supplementation (L-arginine, L-leucine, L-tryptophan), nucleotide supplementation (adenine), and oxygen depletion are described in greater detail enumerating specific MinSpan pathway changes (see Supplementary Information).

**MinSpan pathways aid in experimental discovery of novel regulation**

MinSpan pathways not only accurately predict known TF activities but also offer an opportunity to discover novel regulation. We chose three novel TF–environment predictions to experimentally validate that are non-obvious, in the sense that little to no literature links the TF with the predicted associated environment. To be rigorous in the experimental design, we chose environmental shifts that have been well-studied; where discovering novel experimental findings would be more difficult.

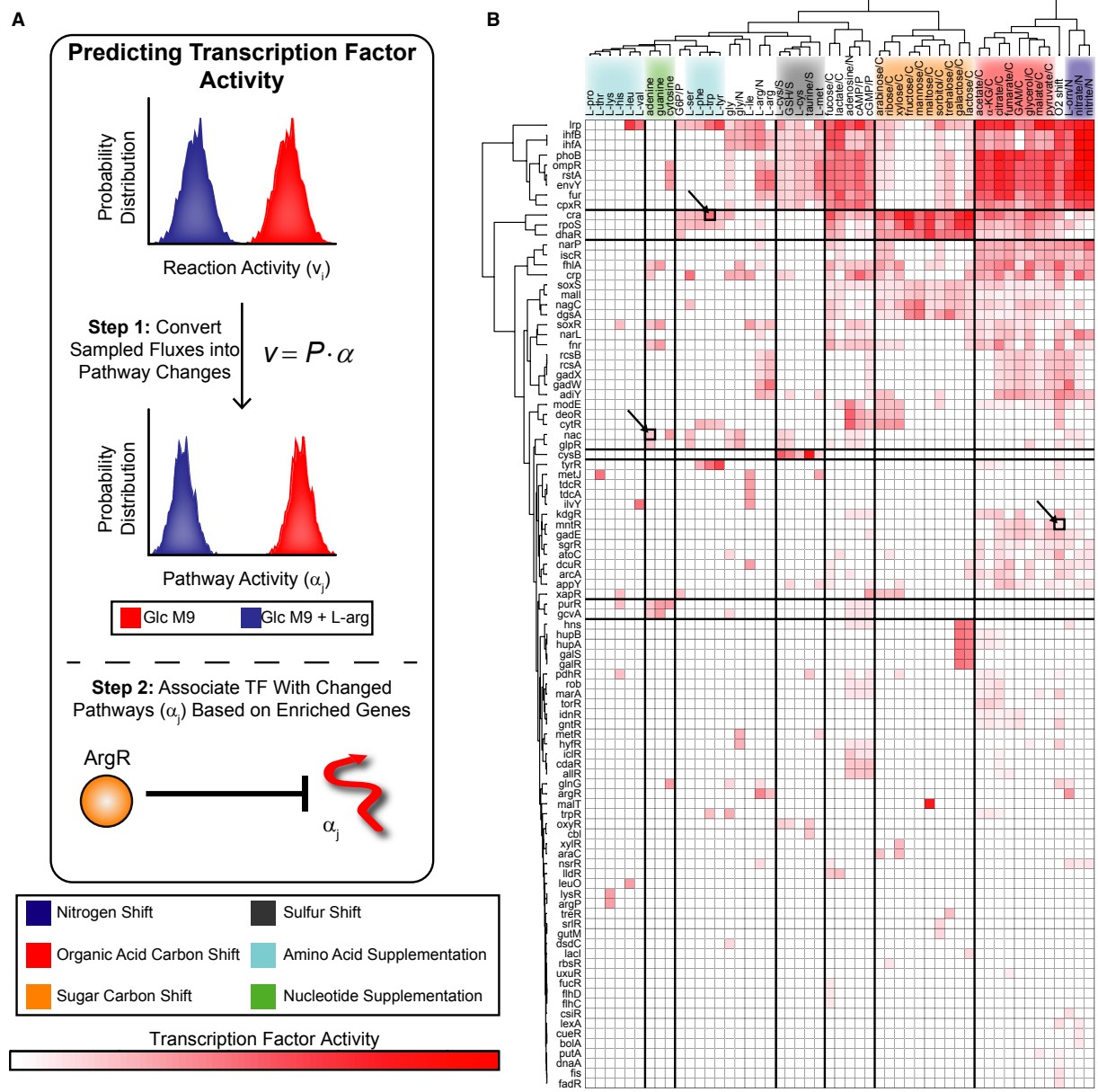

**Figure 5. MinSpan pathways help predict transcription factor activity.**

A  Constraint-based models can determine reaction activity, or flux states (**v**), using Monte Carlo sampling. Decomposing sampled flux states into linear weightings of MinSpan pathways (**α**) allows prediction of TF activity. For example, metabolic reaction fluxes are sampled under glucose minimal media and glucose minimal media + L-arginine supplementation. Typical analysis would yield a list of reactions (including $v_i$) that are significantly changed. With MinSpan pathways, the flux distributions can be converted into significant changes in pathway activity (including $\alpha_j$). TFs are associated with pathways based on enrichment of regulated genes. Predicting TF activity is based on which TFs are associated with the significantly changed pathways; in this case, $\alpha_j$ is associated with ArgR.

B  The TF activity of 51 nutrient shifts was predicted and can be hierarchically clustered by nutrient shift type. TF activity for the heatmap is defined as the percentage of differential MinSpan pathways that are associated with that TF. 36% of the TF–environment associations predicted are not known, providing numerous predictions for experimentation. Experimentally tested TF–environment associations are highlighted.

The three tested associations were Nac with adenine supplementation (ade/Nac), Cra with L-tryptophan supplementation (L-trp/Cra), and MntR with shift to anaerobic conditions (O₂/MntR). The chosen shifts represent three distinct magnitudes of dual perturbations. In the ade/Nac case, the environment and genetic perturbations are both relatively minor. In the L-trp/Cra case, Cra is a broad acting TF and dominates, while the environmental perturbation to the absence of oxygen dominates in the O₂/MntR case.

We generated RNA-seq data from dual perturbation experiments (Ideker *et al*, 2001; Covert *et al*, 2004) for the three cases consisting of perturbations in the environment (media supplementation) and genetics (TF knockout) of *E. coli*. For each case, we determined the

gene set that is exclusively differentially expressed because of the combination of the genetic perturbation and environmental shift (see Materials and Methods) in order to analyze whether the TF plays a role in the environmental shift.

Global analysis of the gene sets, based on enrichment of regulatory interactions with known TF associations (Gama-Castro *et al*, 2011), suggested that predictions for ade/Nac and O₂/MntR were correct, and the L-trp/Cra prediction was indeterminate. In the ade/Nac case, the gene set was enriched with genes known to be regulated by the TFs GcvA, Lrp, and PurR ($P$ = 9.5e-6, 1.6e-4, 1.8e-4, hypergeometric test), suggesting that Nac (nitrogen assimilation control) regulates similar genes during the shift or even regulates the corresponding TFs. In the L-trp/Cra case, there were no enriched TFs suggesting no global consensus. This discrepancy might be due to (1) Cra knockout causing a large genetic shift that might have changed how *E. coli* responds to L-trp and (2) MinSpan is inaccurate for predicting global regulation. In the O₂/MntR case, TFs known to be associated with the anaerobic shift (including ArcA and Fnr) were enriched as a whole ($P$ = 3.6e-3, hypergeometric test).

Through differential expression and detection of high confidence binding sequence motifs, we identified novel regulatory roles for all three tested TFs (Fig 6). For the ade/Nac case, we identified potential regulation of genes involved in purine metabolism, involved in nitrogen assimilation, and regulated by Lrp (Fig 6A). The transcription units (TUs) gcvTHP and gcvB are known to be regulated by GcvA and PurR and are potentially regulated by Nac. Nac also seems to regulate gcvB, which is a small regulatory RNA of Lrp (Modi *et al*, 2011). Using FIMO (Grant *et al*, 2011), we detected a significant Nac binding sequence motif for gcvB ($-173$ bp of transcription start site (TSS), $P$ = 1.73e-6). A significant increase in gcvB suggests a repression of Lrp and genes typically repressed by Lrp should have higher expression and genes activated by Lrp should have lower expression. This trend was observed in all significantly changed expression of Lrp-regulated genes (Fig 6A). We also identified novel regulation and high confidence binding sequence motifs for nitrogen assimilation genes: nirB ($-87$ bp of TSS, $P$ = 7.63e-5) and nrdHIEF ($-170$ bp of TSS, $P$ = 2.2e-5).

Though there was no global trend for the L-trp/Cra case, we did find that Cra potentially regulates the L-trp symporter and tryptophanase (tnaCAB, Fig 6B). Crp is a known regulator of this TU (Botsford & DeMoss, 1971), but our data suggest that Cra also plays a role, possibly by in-direct regulation through Crp (Shimada *et al*, 2011).

Finally in the O₂/MntR case, MntR potentially regulates four TUs highly regulated during the anaerobic shift including the TF GadX (-120 bp of TSS, $P$ = 2.6e-5) (Fig 6C). GadX regulates pH-inducing genes and the GAD system that play roles during fermentation (Tramonti *et al*, 2002). MntR also potentially regulates a subunit of pyruvate formate lyase (yfiD), NADH:ubiquinone oxidoreductase II (ndh), and molybdenum biosynthesis (moaABC).

It is important to note that potential regulatory sites were only detected during dual perturbation, suggesting that experiments to elucidate TRNs under one environmental condition underestimate non-intuitive regulatory events. The additional potential binding sites nearly double the known potential binding sites for Nac and MntR (Keseler *et al*, 2011). Dual perturbation predictions were more accurate for local TFs (Nac/MntR) than global TFs (Cra), which is

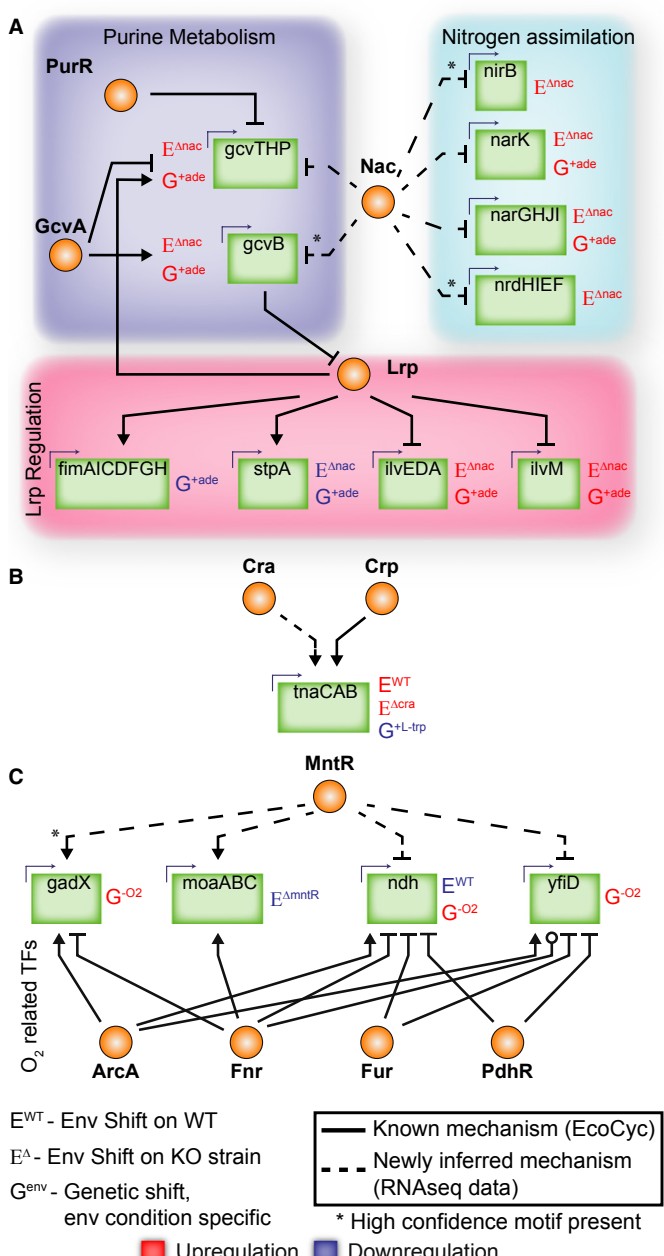

**Figure 6.  MinSpan TRN predictions suggest informative dual perturbation experiments that led to discovery of novel *Escherichia coli* transcriptional regulation.**

A   Nac plays a role in purine metabolism and a larger role in nitrogen metabolism than previously known. Nac also regulates Lrp through gcvB.

B   Cra regulates tnaCAB, which is also subject to Crp regulation.

C   MntR plays a regulatory role for four genes that are heavily regulated by ArcA/Fnr/Fur/PdhR and are utilized during anaerobic conditions.

also consistent with the correlation analysis. The analysis presented to predict TF–environment associations is only possible with MinSpan pathways, as opposed to pathway databases, as MinSpan pathways directly link flux simulations from constraint-based models to pathway biomolecular properties.

## Discussion

High-throughput technologies have transformed biological data generation and experimentation. However, a major remaining challenge is analysis and interpretation of large datasets for achieving biological understanding (Palsson & Zengler, 2010; Sboner *et al*, 2011). Though data analysis is steadily improving, the underlying interpretation is still often relying upon historically determined, human-defined pathways (Khatri *et al*, 2012). In this study, we introduce an unbiased genome-scale method to define pathways based on whole network function and a principle of parsimonious use of cellular components. We find that the MinSpan pathways are not only biologically relevant in their ability to recapitulate independent datasets on biomolecular interaction, but are surprisingly more accurate than traditional pathway databases such as KEGG, EcoCyc, YeastCyc, and Gene Ontology. The results have three implications.

First, the results suggest that the traditional approach to defining metabolic pathways is not complete and an unbiased alternative might be more representative of the underlying pathway structure. Traditional pathway enumeration focuses predominantly on biochemical reactions. By incorporating a minimal criterion of the number of reactions used, the MinSpan approach indirectly introduces the requirements for biomolecular machinery usage into the pathway definition. There are two fundamental features of MinSpan pathways that differentiate them from traditional pathways. (1) MinSpan pathways account for all necessary components to make a pathway fully functionally independent (e.g., they are network-based). (2) MinSpan pathways represent the simplest pathway structure in a given network context. The improved consistency of MinSpan with biomolecular interactions suggests that the coordinated regulation and usage of biological components in the cell have evolved to be minimal and independent in order to adapt to perturbation with as little cost to the cell as possible. Biologically meaningful pathways may be minimal, independent, and segregated. Future delineation of metabolic pathways, both network and human-defined, should take into consideration the cost of biomolecular machinery and systemic functional requirements for metabolic function.

Second, the MinSpan pathways provide an alternative, complementary, and potentially more powerful approach for investigators to analyze their generated data. Current pathway databases are tremendously important in conceptualizing biological function and are used by numerous investigators for data analysis. As MinSpan pathways are more biologically relevant in terms of the underlying biomolecular interactions, the theory presented here for an unbiased pathway structure opens up the potential for a whole new suite of pathways to be used with tools such as Gene Set Enrichment Analysis (GSEA) (Subramanian *et al*, 2005).

Third, MinSpan pathways can guide the difficult process of reconstructing and determining TRNs. The best characterized TRN is that in *E. coli*. Though the *E. coli* TRN is not complete, our approach of coupling metabolic models with MinSpan pathways identifies novel associations between TFs and environmental shifts, providing a rational method to design context-specific dual perturbation experiments. In this study, we have experimentally validated three of the 93 novel predictions allowing us to double the known potential regulatory sites for Nac and MntR. Our analysis shows that experiments under varying environmental conditions are required to elucidate novel regulatory roles. Further, we recently confirmed two

additional novel MinSpan predictions for cytosine/Nac and cytosine/NtrC (Kim, 2014) using chromatin immunoprecipitation with exome sequencing methods (Rhee & Pugh, 2011). This work also confirms the biochemical binding of Nac to the transcription unit gcvTHP, which is the mechanism for which Nac is involved in purine metabolism (Fig 6A). The remaining 88 MinSpan predictions provide a roadmap for future experimentation to help discover numerous new regulatory roles in *E. coli*, and the overall method can be applied to any organism with a metabolic and regulatory network.

There are also some limitations and areas for further research with regards to the MinSpan algorithm. First, the MinSpan algorithm is dependent on the quality of the genome-scale metabolic model utilized; in the same way, the quality of pathway databases for particular organisms is dependent on the biochemical knowledge available. The MinSpan pathways are more comprehensive in *E. coli* than *S. cerevisiae* as iJO1366 is much more complete than iMM904. This difference is a reflection of the biochemistry of *E. coli* being better studied than *S. cerevisiae*. Second, just as there are multiple pathway databases for the same organism, there are sometimes multiple metabolic network reconstructions for the same organism. Further research is needed to assess the differences in the calculated MinSpan pathways for different metabolic models of the same organism. Third, human-defined pathways are often defined in a universal, rather than organism-specific, manner. This can be a strength, particularly for educational purposes as it provides a common "language" to describe the function of many organisms. However, universal pathways can also be a weakness. Human-defined pathways focus on the topology of gene products, while ignoring the organism-specific functional context of metabolic pathways. For example, isotopomer metabolic flux profiling has shown that metabolic functionality can often be quite different than the gene products present (Amador-Noquez *et al*, 2010). To assess the conservation of MinSpan pathways, a preliminary analysis comparing *E. coli* and *S. cerevisiae* MinSpan pathways is provided in the Supplementary Information. However, further research with several reconstructions of different organisms is needed to fully assess whether MinSpan pathways are conserved across species.

## Materials and Methods

### MinSpan Formulation

The MinSpan algorithm determines the shortest, linearly independent pathways for a stoichiometric matrix (**S**) with dimensions $m \times n$ and a rank of $r$. The input is a metabolic model (variables $S$, lb, ub) and outputs a MinSpan pathway matrix (**P**). **P** is the sparsest null space of **S** that maintains biological and thermodynamic constraints (lb and ub). Coleman and Pothen defined the mathematical problem for the sparsest linear basis of the null space as the "sparse null space basis problem" and then proved that a greedy algorithm must find the globally optimal sparsest null space (least number of non-zero entries) (Coleman & Pothen, 1986). More recently, Gottlieb and Neylon showed that a similar problem, "the matrix sparsification problem," is equivalent to the "sparse null space basis problem" (Gottlieb & Neylon, 2010). We formulated "the matrix sparsification problem" as a mixed-integer linear programming

(MILP) problem. The MILP is boxed in the pseudo-code below. Simply put, the orthonormal null space (**N**) of **S** is initially defined by singular value decomposition. Then, the vectors of the orthonormal null space are iteratively replaced by the shortest pathways that span the removed vector's subspace. This process is continuously repeated until the number of non-zero entries in **P** has converged to a minimum. Before running the algorithm, all reactions that cannot carry a flux are removed, all exchanges are opened, and the biomass function is removed. The algorithm is summarized below:

$$\mathbf{N} = \text{null}(\mathbf{S})$$

$$\mathbf{P} = \mathbf{N}$$
while (true)
　　$\mathbf{P^0} = \mathbf{P}$
　　for j = 1:n-r
　　　$\mathbf{P'_{a,b}} = \begin{cases} \mathbf{P_{a,b}} & \text{if } b \neq j \\ \mathbf{0} & \text{if } b = j \end{cases}$

　　　$\mathbf{x} = \mathbf{N} \cdot \text{null}(\theta^{\mathrm{T}})$　where $\mathbf{N} \cdot \theta = \mathbf{P}$

　　　$$\boxed{\begin{array}{l} \min \sum b_i \text{ where } b \in \{0,1\} \\ \mathbf{S} \cdot \mathbf{v} = 0 \\ lb_i \leq v_i \leq ub_i \\ -1000b_i \leq v_i \leq 1000b_i \\ \mathbf{x}^{\mathrm{T}} \cdot \mathbf{v} \neq 0 \end{array}}$$

　　　$\mathbf{P'_{a,b}} = \begin{cases} \mathbf{P_{a,b}} & \text{if } b \neq j \\ \mathbf{v} & \text{if } b = j \end{cases}$

　　end
　　$\mathbf{P} = \mathbf{P'}$
　　if nnz(**P**) == nnz(**P⁰**), break, end
end

The null() operator defines the orthonormal null space using singular value decomposition. The nnz() operator determines the number of non-zero entries in a matrix. Vectors are in bold, and matrices are capitalized. **P'** is similar to **P**, but without the vector **p_j**, and **x** is a vector that spans the space of **p_j** and is linearly independent from **P'**. **b** is a binary version of the flux vector (**v**) that is minimized by the optimization problem to determine the MinSpan pathways. This is proved by contradiction. If **P'** and **x** are linearly dependent, then multiples of **x** and the vectors of **P'** should linearly combine to zero:

$$\mathbf{P'}\Lambda + \mathbf{x}\lambda = 0 \text{ where } \Lambda, \lambda \neq 0$$
$$\mathbf{P'}\Lambda = -\mathbf{N} \cdot \text{null}(\theta^{\mathbf{T}})\lambda$$
$$\mathbf{N}^{-1}\mathbf{P'}\Lambda = -\text{null}(\theta^{\mathbf{T}})\lambda$$
$$\theta\Lambda = -\text{null}(\theta^{\mathbf{T}})\lambda$$
$$\theta^{\mathbf{T}}\theta\Lambda = -\theta^{\mathbf{T}}\text{null}(\theta^{\mathbf{T}})\lambda$$
$$\theta^{\mathbf{T}}\theta = 0$$

$\theta^{\mathrm{T}}\theta$ is a positive semi-definite matrix by definition and cannot equal zero. Thus, **P** and **x** are linearly independent. The $\mathbf{x}^{\mathrm{T}} \cdot \mathbf{v} \neq 0$ constraint ensures that the calculated pathway spans the proper dimension of the null space. For a MILP problem, the constraint is formulated as below, where $\varepsilon$ is an arbitrarily small value (set to 0.1, various other choices yield similar results), and

$f^+$ and $f^-$ are binary variables required to formulate a "not-equal" constraint:

$$f^+ + f^- = 1 \text{ where } f \in \{0,1\}$$
$$\mathbf{x}^{\mathrm{T}} \cdot \mathbf{v} - 1000(1 - f^+) \geq \varepsilon f^+$$
$$-\mathbf{x}^{\mathrm{T}} \cdot \mathbf{v} + 1000(1 - f^-) \geq \varepsilon f^-$$

The termination criterion for the branch and bound method for each MILP iteration is a relative gap tolerance of 1e-3 or time limit of 30 min. These criteria were developed based on convergent properties of solutions as the two parameters were varied. As the MinSpan algorithm is a MILP problem, there can exist alternative optimal solutions. The total number of non-zero entries in the matrix is unique, but the pathways may not be. Rerunning the correlation analysis to determine biological relevance with alternate MinSpan pathways for both *E. coli* and *S. cerevisiae* yielded little changes to the results (see Supplementary Information). MinSpan is available for COBRApy at https://github.com/sbrg/minspan.

MaxSpan and RandSpan were generated similarly with slight modifications. For MaxSpan, the optimization was switched to a maximization problem to make the densest null space. For RandSpan, we assigned a random value from −0.5 to 0.5 to the optimization **c** vector to randomly minimize and maximize the use of reactions while constructing each null space. For maximization, the following constraints are added to link the binary and continuous variables:

$$v_i - b_i \geq 1000(g_i - 1) - (1000 - \varepsilon)$$
$$v_i + b_i \geq -1000g_i - (1000 - \varepsilon)$$

Where $\varepsilon$ is an arbitrarily small value (set to 0.1, various other choices yield similar results) and $g_i$ are dummy binary variables to allow an "OR" statement in linear programming. If $g_i$ is either 1 or 0, one of the two above constraints is off.

## Correlation analysis

The metabolic reactions in MinSpan pathways were converted to gene and protein sets based on the gene-protein-reaction associations. Pairwise Spearman rank correlations of the co-occurrence or co-absence of genes and proteins across the pathways were calculated. Correlations that were not significant ($P > 0.05$, permutation test) were filtered from further analysis. The total number of correlations remaining after filtering is the coverage criteria (i.e., log (interactions)) in the x-axes of Fig 2. The correlation coefficient was the varied discrimination threshold for generating the receiver operating characteristics (ROC) curve, using a convex hull. P-values to determine statistically different ROC convex hull curves were calculated using the approach by Hanley and McNeil (Hanley & McNeil, 1982).

The correlations were compared to biomolecular interaction data types to assess the biological relevance of gene and protein groupings of the pathways. Known biomolecular interactions were taken as the gold standard positive set. All available yeast two-hybrid screening data from BioGRID (Stark *et al*, 2006) were used for protein-protein interactions. "Stringent" positive genetic interactions (defined by the authors as $P < 0.05$ and $\varepsilon > 0.16$) were taken from

Costanzo *et al.* For TF–gene interactions, all reported interactions in RegulonDB (Gama-Castro *et al*, 2011) were used. A lack of reported biomolecular interactions in these three datasets was deemed the gold standard negative set.

KEGG modules (Kanehisa *et al*, 2012) and GO Biological Processes ontology (Ashburner *et al*, 2000) were downloaded from their respective websites on 01/27/2013 for comparison. EcoCyc (Keseler *et al*, 2011) and YeastCyc (Cherry *et al*, 2012) pathways were downloaded on 02/12/2013. Only distinct pathways with two or more genes from the metabolic networks were considered. The same correlation analysis was used for the human-curated pathways, MaxSpan, and RandSpan.

### Comparing pathway databases

We calculated the connection specificity index (CSI) (Green *et al*, 2011; Bass *et al*, 2013) between pathways based on their gene products across all pathway definitions (MinSpan, KEGG, BioCyc, and Gene Ontology). The CSI is a metric that determines the similarity between two vectors by ranking the Pearson's correlation coefficient of the two vectors based on the correlations of all other vectors versus the two vectors in question. A CSI between pathways A and B is defined as:

$$CSI_{AB} = \frac{\# \text{ pathways correlated with A and B that } PCC < PCC_{AB} - t}{n_y}$$

where PCC is the correlation coefficient of gene products, $PCC_{AB}$ is the correlation between A and B, $t$ is an empirically derived threshold, and $n_y$ is the total number of pathways. Further explanation of CSI and software tools for its use is available (Bass *et al*, 2013). The threshold for CSI was set based on the distribution of correlations (*E. coli* – 0.0350, *S. cerevisiae* – 0.0562, see Supplementary Figure S13). Pathways were considered similar if their CSI ranked in the top 15% of CSI values.

We also employed a k-nearest neighbors search to find the most similar pathways across the pathway databases. The Pearson's correlation was used as the distance metric. The closest hit that also had a high CSI value (top 15%) was used as the nearest neighbor. If the pathway did not have a high CSI value, then the pathway was deemed unique compared to the other databases.

MinSpan pathways contain many gene products related to the mass balancing of the network, such as transporters, that are active in nearly every pathway. In order for a meaningful comparison between MinSpan and human-defined databases, the genes in each MinSpan pathway were filtered to only the representative genes of that pathway. To do so, we used a conservative filter to remove genes that were in nearly every pathway ($P > 0.85$, hypergeometric, empirically derived).

### Determining reaction fluxes and transcription factor activities

Monte Carlo sampling (Schellenberger *et al*, 2011) was used to determine 10,000 reaction flux distributions for the *E. coli* metabolic network for each of the 52 nutrient conditions. For glucose minimal media conditions, exchange constraints were taken from Covert *et al* (2004). For anaerobic conditions, rate of oxygen input was set

to zero. For amino acid and nucleotide simulations, rate of amino acid or nucleotide input was set to the minimum rate that would allow the biomass constituent, at the wild-type growth rate, to be generated solely from exogenous substrate. Six nutrient shifts were not considered (L-alanine, L-asparagine, L-aspartate, L-glutamate, L-glutamine, and uracil), as they are involved with type 3 loops (physiologically infeasible pathways that are artifacts of pathway enumeration algorithms (Palsson, 2006)). For alternate carbon, nitrogen, phosphorus, and sulfur sources, the original input source from glucose minimal media conditions was set to zero and an equal rate of atom flow was set as the input. All nutrient conditions had the basal biomass flux rate set to 90% of the optimal value. To compare different conditions, sampled fluxes within a particular condition were normalized by median growth rate for comparison. The specific lower bounds for exchanges used with the *E. coli* model are detailed in the Supplementary Dataset S4. All upper bounds are set to 1,000.

Assignment of TFs to MinSpan pathways was by hypergeometric enrichment ($P < 0.01$) of the TF-regulated genes as determined by RegulonDB (Gama-Castro *et al*, 2011). Determination of whether or not a TF plays a role in an environmental shift was determined by hypergeometric enrichment ($P < 0.05$) of the number of significantly changed TF-associated pathways. TFs regulating only one metabolic reaction or appearing in one pathway were removed due to a lack of statistical power.

### Growth conditions, RNA isolation, and RNA-seq

KEIO collection knockout strains were used (Baba *et al*, 2006). Strains were grown to mid-exponential phase under conditions specified in Supplementary Table S1. Spinner flasks were used for aerobic culture and serum bottles for anaerobic cultures. One volume of mid-exponential sample was mixed with two volumes of RNA-Protect (Qiagen). Cell pellets were lysed for 30 min at 37°C using Readylyse Lysozyme (Epicentre), SuperaseIn (Ambion), Proteinase K (Invitrogen), and SDS. Following cell lysis, total RNA was isolated using RNeasy columns (Qiagen) following vendor procedures with on column DNase treatment for 30 min at room temperature. Paired-end, strand-specific RNAseq was performed using the dUTP method (Levin *et al*, 2010) with the following modifications. rRNA was removed with Epicentre's Ribo-Zero rRNA Removal Kit. Subtracted RNA was fragmented for 3 min using Ambion's RNA Fragmentation Reagents. cDNA was generated using Invitrogen's SuperScript III First-Strand Synthesis protocol with random hexamer priming.

### Transcript quantification from RNA-seq reads

RNA-seq reads were aligned to the genome sequence of *E. coli* (RefSeq: NC_000913) using Bowtie (Langmead *et al*, 2009) with 2 mismatches allowed per read alignment. To estimate transcript abundances, FPKM values were calculated using Cufflinks (Trapnell *et al*, 2010) with appropriate parameters set for the strand-specific library-type and upper-quartile normalization. EcoCyc annotations were used for transcript quantification. Differential expression analysis was done using cuffdiff with upper-quartile normalization, and appropriate parameters set for strand-specific library type. A fold change of greater than 2-fold and a false discovery rate

cutoff of 0.05 were used to determine significant differential expression.

### Analysis workflow for dual perturbations

Dual perturbation experiments consisted of four RNA-seq experiments: (1) wild type (WT) on glucose minimal media, (2) WT + nutrient shift, (3) TF knockout (KO) on glucose minimal media, and (4) TF KO + nutrient shift. We defined differentially expressed gene sets between the four conditions as: E1 (WT versus WT + nutrient), E2 (KO versus KO + nutrient), G1 (WT versus KO), and G2 (WT + nutrient versus KO + nutrient). The differential gene set of the combination of environmental and genetic perturbations is defined as the union of the following sets: $\{\{E2 \triangle E1\}\backslash G1$ and $\{G2 \backslash G1\}$. The gene sets for the three experiments are provided in Supplementary Dataset S5. To globally determine prediction accuracy, we used RegulonDB to determine whether a gene set was enriched ($P < 0.05$, hypergeometric test, Bonferroni correction) in genes of a particular transcription factor. A prediction was deemed correct if the enriched transcription factors had known associations with the environmental shift.

**Supplementary information for** this article is available online: http://msb.embopress.org

### Acknowledgements

We would like to thank Gabriela Guzman for aiding HL with experiments and Dr. Niko Sonnenschein, Dr. Neema Jamshidi, and Daniel Zielinski for valuable discussions. This work was supported by NIH grant GM068837 and the Novo Nordisk Foundation. HL is supported through the National Science Foundation Graduate Research Fellowship (DGE1144086). RNA-seq data are available at Gene Expression Omnibus (GSE48324).

### Author contributions

AB, JS, and AE developed the MinSpan algorithm. HN analyzed the RNA-seq data. HL generated all experimental data. SF provided expert insight into the *E. coli* TRN. AB completed all other analyses and wrote the manuscript. AB, HN, NEL, and BOP designed the study. All authors edited and approved the final manuscript.

### Conflict of interest

The authors declare that they have no conflict of interest.

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
