## [Review Process File · Molecular Systems Biology]

Minimal metabolic pathway structure is consistent with associated biomolecular interactions

Aarash Bordbar, Harish Nagarajan, Nathan E. Lewis, Haythem Latif, Ali Ebrahim, Stephen Federowicz, Jan Schellenberger, Bernhard O. Palsson

Corresponding author: Bernhard O. Palsson, University of California San Diego

Review timeline:	Submission date:	02 August 2013
	Editorial Decision:	09 October 2013
	Appeal:	14 October 2013
	Editorial Decision:	22 October 2013
	Resubmission:	28 February 2014
	Editorial Decision:	24 April 2014
	Revision received:	20 May 2014
	Accepted:	26 May 2014

Editor: Thomas Lemberger/Maria Polychronidou

Transaction Report:

1st Editorial Decision

09 October 2013

Thank you again for submitting your work to Molecular Systems Biology. First of all, I would like to apologize for the exceptional delay in getting back to you. Unfortunately it took a considerably long time until we received the report of referee #3. We have now heard back from the four referees who agreed to evaluate your manuscript. As you will see from the reports below, the referees raise substantial concerns on your work, which, I am afraid to say, preclude its publication in Molecular Systems Biology.

Overall, the reviewers acknowledge that the presented 'minimal pathway'-based approach is potentially interesting and reviewers #2 and #3 are cautiously positive. However, reviewers #1 and #4 raise significant concerns regarding the biological significance of the MinSpan pathways. In particular, they feel that as it stands, the study remains too preliminary and provides only limited biological support for the physiological relevance of the MinSpan pathways and their superiority compared to conventional pathways.

Under these circumstances, I see no other choice than to return the manuscript with the message that we cannot offer to publish it. In any case, thank you for the opportunity to examine your work. I hope that the points raised in the reports will prove useful to you and that you will not be discouraged from submitting future work to Molecular Systems Biology.

<http://www.nature.com/msb>

Reviewer #1 (Report):

This paper addresses an important issue, namely how to define biologically meaningful metabolic pathways in an unbiased manner. The MinSpan pathways potentially fulfil this criterium. I have some major concerns however. As it stands it is difficult to grasp the essence of the MinSpan pathways and I miss a large part of the biological support which the authors claim to have. This is at least partly due to a superficial and inaccurate way of formulating and partly because of a lack of data and/or unclear presentation of the data.

Specific comments

In the introduction the authors address the limited applicability of convex-based pathway definitions, such as EFMs, due to combinatorial explosion. This combinatorial explosion has been analysed and brought back to multiple pathways in just a small set of subnetworks (DOI: 10.1038/srep00580). Could the authors comment on this work and relate it to the MinSpan pathways?

The MinSpan algorithm is exactly described for computational systems biologists, but an insightful interpretation for a wider systems biology audience is lacking. The small example gives limited insight due to the oversimplification of the pathway (Fig 1). All reactions are unimolecular, while real metabolic networks are highly connected via cofactors such as ATP or NAD(P)H. The simplified TCA cycle cannot carry any flux at steady state. Unlike the real TCA cycle it has no net input or output and can therefore only evolve to equilibrium. It is unclear why the same reaction is sometimes represented as irreversible and sometimes as reversible in the decomposition shown in Fig 1. The comparison to the EFMs is highly relevant and should better be included in the main text rather than in the supplement (Fig S1 and S2). The explanatory power of this paper can be greatly improved by the use of a more realistic example, including interconnections via a cofactor, and by a more exact definition of the MinSpan pathways.

With respect to Figure S2 above: From the legend I have no clue how this represents a comparison between MinSpan and extreme pathways. Please, add a descriptive legend explaining what is exactly presented.

Biological support comes from different kinds of interaction networks. I miss an exact explanation as to why these networks support the MinSpan pathway definition. For TF networks this is rather intuitive, but protein-protein interactions are surprising. Metabolites diffuse from one enzyme to the next and channeling has been demonstrated only in rare cases. Hence a physical interaction, as measured by the 2-hybrids, is not essential for proper functioning of a pathway.

The flux calculations with the Cobra toolbox are hardly described. Were any modifications, even small updates, made to the cited models? If so, SBML files should be included. Which 52 nutrient conditions were applied and which constraints were imposed? Are there any particular settings in the algorithm that influence the outcome? Please, give an explanation that enables a qualified scientist to repeat the work.

With respect to figure 4 where calculated TF activity is correlated to calculated flux, it is claimed that 37 out of 51 nutrient shifts matched known associations (true positives). What is the exact criterium for a true positive, i.e. what is considered minimum information to call an association a true positive? I have been looking for a list of associations and the proof for them being true/false or undetermined, but I did not find it. In the supplement I only find a general list of papers cited, but no quantitative analysis of the results.

The authors report that global analysis of the gene sets suggested that the predictions for *ade/Nac* and *O2/MntR* were correct, but I cannot evaluate the validity of this statements. To this end RNAseq datasets were generated. However, in the Materials and Methods only limited information is given about the experimental setup and the results are not shown.

Throughout the paper, I often wonder what a sentence means exactly. The paper could be improved substantially by checking the clarity of each statement. This also holds for figure legends, which give conclusions but often lack sufficient information to understand what is precisely depicted

in the figure.

Reviewer #2 (Report):

In their work, Bordbar et al. propose a new definition of metabolic pathways which are derived computationally in a maximum parsimony framework from a genome-scale metabolic network. This addresses an important problem in metabolic research, namely the decomposition of a metabolic network in minimally functioning units. While traditionally classically defined metabolic pathways have been used for this purpose, these type of metabolic pathways often hamper new discoveries e.g. if pathways usually not considered together in the classical context are interacting. While approaches such as elementary flux modes and extreme pathways have been proposed to overcome these limitations, they are not applicable to genome-scale metabolic networks. In this context, the approach presented by the authors represents an important step for a novel automated definition of metabolic pathways. They demonstrate that their pathways are more representative than classically defined pathways based on the analysis of interactions in several large-scale datasets. However, this problem might be remedied by alternative approaches such as partially coupled reaction sets. There are only a couple of issues I would like the authors to address relating to the uniqueness of the MinSpan pathways and the representation of metabolic pathways.

Major issues:

It is not entirely clear to me how "stable" the MinSpan pathways are. Since they are based on a MILP-formulation, alternative solutions might exist. Moreover, small modifications in the stoichiometric matrix such as adding or leaving out a reaction might give rise to entirely different MinSpan pathways. Thus, the authors should discuss whether they will always find very similar pathways even if they "perturb" the metabolic network. Also they should analyse whether alternative optima of the MILP-approach exist and to which extent these optima differ from each other.

Also along these lines I would be interested in "false negatives", that is, to which extent classically defined pathways (e.g. in EcoCyc) are not reflected by the MinSpan pathways.

The approach presented by the authors should be made available, e.g. as component of the COBRA toolbox.

Minor issues:

There appears to be an error in the RNAseq group assignment: there are two groups (E2 and G2) with the same conditions.

Though the shortest path between two major metabolites can often be the most accurate representation, it does not have to be (p. 18)

No explanation of "type 3 loop" (p. 23)

A accession number for the RNAseq data should be provided.

Reviewer #3 (Report):

Bordbar et al. present an interesting new approach for the unbiased definition of metabolic pathways. The new method scales linearly with the size of metabolic networks, allowing for a structural definition of pathways that can sustain fluxes at steady-state on a genome scale. The predicted pathways correlate well with data on protein and gene interaction networks, and predict potential gene-transcription factor associations. I think this is an interesting contribution and probably deserves publication.

Main comments:

1. Although the authors provide a description of global pathway properties in figure S3, I feel it is important to provide this or similar description in the main text. In particular, the typical pathway sizes and the number of pathways per reaction should be discussed. The method enumerates a large number of pathways in both organisms tested, but one immediate question is whether these pathways have some clear biological characteristics or if there are many spurious pathways that would be hard to interpret.

2. While the authors point out that historically biochemical pathways have been defined in a universal fashion rather than being organism specific, there is no clear reference as to how minSpan pathways discovered in two organisms compare to each other. One reason classical biological pathways are popular is that they are generally consistent between species. Are minSpan pathways well or poorly conserved in multiple species?

3. The authors attribute the larger number of pathways in *E. coli* to a more complete metabolic network, implying that the definition of minSpan pathways depends on the level of curation and pre-existence of genome-scale metabolic models. How sensitive are minSpan pathways to the level of curation in different models? Even for the same organism, different reconstructions are published by different groups, and different models are of varying quality. These limitations should be discussed.

Reviewer #4 (Report):

I have already reviewed this manuscript when it was submitted to a different journal. As major concerns remain unaddressed, I copy-paste here my last recommendation:

Our existing biochemical pathways for metabolism are human-made and they are thus not necessarily a representation of the actual underlying biological function. Here, (i) the authors develop a computational algorithm, with which they can extract functional units/pathways from genome-scale metabolic networks, (ii) they argue that these extracted MinSpan pathway correlate better with data on protein-protein interactions, transcriptional networks, and genetic interactions than human-made pathways do and (iii) they use the new pathways to identify few new transcription factor interaction sites. While I see some value in their work, I feel that this manuscript is more suitable for a specialized journal. Furthermore, I have concerns about the general validity of their claims.

1. Importance? Relevance?

"Pathways" are required for biochemical education and, for instance, for the analysis of experimental (omics-) data (to provide "context" for the analysis). While the new MinSpan pathway definition seems to have some advantages over the human-defined pathways (e.g. as shown in Figure 3A, B), a number of questions arose, which make me doubt that the new pathway concept will have a widespread impact:

- a) It is not clear how "reasonable" the pathways are that the authors find. They mention some "reasonable" ones, but what about the rest? What is the degree of agreement between the human-defined ones and the new ones? Do we get back the TCA cycle, glycolysis, and pentose phosphate pathway?
- b) In any case, I have very strong doubts that our biochemical textbooks are going to be rewritten; further I doubt that the widely used resources such as KEGG, BioCyc, etc. are going to replace the human-made pathways by the MinSpan pathways.
- c) If this doesn't happen, then the only point that is left from the authors' work is a tool to computationally define new pathways that can then be used to analyze, e.g. omics, data, where we so far use human-defined pathways. However, it is not clear whether the new pathways are so much more powerful as compared to the old ones.

Overall, while I see some value in the authors work (because the new pathway concept could make one or the other computational algorithm a bit better), I feel that the work is not groundbreaking enough to be published in MSB. I feel that the work is more appropriate for a specialized journal.

2. Claims fully supported?

The authors claim that their MinSpan pathways reflect the biological reality better than the human-defined pathways. While this is a point that most people would generally be tempted to believe, I am not sure whether the authors present sufficient evidence that their claim is actually true:

- a) Their work rests on the assumption that a functional pathway needs to correlate with PPI, genetic interactions, and co-regulation. However, they do not present any evidence for this assumption. E.g. they state "As most PPIs occur between proteins within the same metabolic complex or adjacent

metabolic reactions ...", but they don't provide a reference for this statement.

b) Even if this assumption is true, I am not sure whether there is sufficient statistical evidence that the MinSpan pathways are better, although this is crucial for their whole story. First, unfortunately, on the basis of the provided information, it is impossible to understand what the authors did to come to the key figures 2D-E. E.g. it is unclear what the y-axis in Figure 2D, E and F mean. The applied statistics are described with only three cryptic sentences on page 23, which are insufficient to allow for solid evaluation of the validity of the claim; e.g. what kind of "filter" was applied? What is the "total number of correlations"?, what is a threshold parameter? Etc. Furthermore, I do not understand that the "random" area is so small and why it sits where it sits. Second, the more illustrative (non-statistical) support that they use to back up their claim (i.e. Figures 3A-C) is nice and helpful, but it is completely unclear whether the shown examples are just a few hand-picked anecdotal examples, which by accident made sense, or whether there would be many, many more of such findings.

3. Problem with the second part:

In the second part, the authors make prediction in reaction fluxes upon environmental/genetic perturbations, then they use MinSpan pathways to provide context for the flux changes and search for enrichment of certain transcription factors in the changed MinSpan pathways. Generally, I have a number of problems with this second part:

- a) The integration with the rest of the text is not optimal; i.e. the story drifts away from MinSpan. In fact, this part reads as if it was written by a different person...
- b) In this second part, the authors need to computationally estimate intracellular fluxes in a number of mutants and under a number of different environmental predictions: Reasonable predictions of changes in reaction fluxes can (assuming that the applied sampling approach can in fact yield reasonable results) only be done if the respective substrate uptake rates are known. First, it is not clear what the authors have precisely done in their simulations (because the text is a bit cryptic, e.g. "For alternate carbon ... the original source was set to zero and an equal amount of atoms was set as the input? Do they mean "amount" or do they mean "rate"? Do they mean *carbon* atoms? What do they mean by "fluxes were normalized by median growth rate"? Where did they get the growth rates from?). Second, whatever they have done, growth rates cannot be a priori predicted by FBA (unless one uses experimentally determined uptake rates; or, if one is interested in flux changes between conditions, if one knows the changed uptake rates between conditions). As I do not see that the authors have done this, it is not clear how they can reliably estimate difference in intracellular fluxes, which is a necessary requirement to determine the flux load of the MinSpan pathways, which in turn was necessary for their statistical analysis shown in Figure 4B.
- c) Generally, this part of the manuscript is not fully clear; for example:
 - a. It is not fully clear how the results shown in Figure 4B were generated (pages 14/15).
 - b. The transcription factor MntR is - according to Ecocyc - associated with 4 proteins (<http://www.ecocyc.org/ECOLI/NEW-IMAGE?type=ENZYME&object=CPLX0-7672>). However, none of these 4 proteins have anything to do with the metabolic changes that I would expect to happen with a shift to anaerobic conditions - which is where the authors would predict that this TF does something. Thus, I wonder how it happened that this TF was found to be connected with MinSpan pathways that would need to change when shifting from aerobic to anaerobic conditions? But eventually, I have missed something
 - c. Further, it is not clear how the authors predicted the new TF-gene interactions mentioned on page 17.
- d) Generally, I guess the authors would like to use this part as a demonstration that the use of their new pathways can support new biological discovery. However, here they followed up on three randomly picked (?)TF-environment interactions and on the basis of these, it is not clear whether they have just been lucky to find new biology. Furthermore, it is not clear whether the same discoveries would not also have been possible with human-made pathways.

Minor comments:

1. How many of such MinSpan pathways exist? How does it look in central metabolism? In linear biosynthesis pathways?
2. The authors argue that a functional unit of a metabolic pathway should have protein interactions. What is the basis for this assumption?
3. Why are the data points of "Biocyc" and "KEGG" in the Figures 2D-F so much different, while the networks in these databases are predominantly the same human-derived ones?
4. Not 100% clear what the nutrient shifts are? Is this always glucose + something? Or only the mentioned nutrients?

5. On page 15, the authors say that 37 out of the 51 nutrient shifts matched known associations ... It is not 100% clear how they got to this statement?
6. While in the text, the transcription factor Cra is called Cra, in Figure 4B the authors use the old term FruR. This is extremely confusing for readers who are not so familiar with *E. coli* transcriptional regulation.
7. In Figure 4B, I would suggest that the authors should highlight the discussed interactions such that the reader doesn't have to search them.
8. Last sentence on page 16: "all three TF": At first, it is not clear, to which three TF are the authors referring to.
9. Are "receiving operating characteristics" not rather called "receiver operating characteristics"?
10. Referencing to the supplementary information is badly done in the text. In fact, the supplementary information reads partly like left over text. For example, I do not know what the authors want to tell in the supplementary section "Determining the 28 transcription factors ..."

Appeal

14 October 2013

I would like to thank you and the reviewers for taking the time to consider our manuscript, titled: "Minimal metabolic pathway structure is consistent with associate biomolecular interactions" (MSB-13-4756). We are encouraged to see that most of the reviewers like the concept of the work and its importance for the systems biology field. However, I believe there has been a misunderstanding on key points of the manuscript that have made two of the reviews negative, particularly concerns of the biological importance of this work.

In this manuscript, we develop an automated pathway framework, define a metric to determine whether the pathways are biologically relevant, and compare automated and human-derived pathways using this metric. The defined metric is based on biomolecular interaction networks. We state that the genes of a metabolic pathway must preferentially conserve transcriptional regulatory mechanisms and positive genetic interactions. This is not a new idea. Many papers have shown that pathways often utilize the same transcription factor or regulatory mechanisms (including Wessely MSB 2010, cited in main text). A positive genetic interaction, by definition, occurs between genes of the same pathway which has been described in numerous papers (including Kelley and Ideker Nat Biotech 2005, cited in main text). In addition, we state that the proteins of a biologically important pathway must preferentially conserve protein-protein interactions. It is well known that interactions between proteins often occur in the same biological function and link cellular processes (Uetz et al. Nature 2000, cited in main text). Looking at this globally for our statistical analysis, we are not stating that every gene or every protein in a pathway must have these interactions, which Reviewer 1 and 4 seem to think is our statement, but overall, there should be a preferential trend. This is our metric for comparing the biological relevance of MinSpan and human-derived pathways. We do not believe that this is a controversial metric.

Using this as our metric, we compared MinSpan pathways conservation of such interactions versus the conservation through human-derived pathways. Figure 2 shows that for 1) three types of interactions, 2) across the two best studied organisms for such interaction networks, and 3) against three different human-derived pathway databases, the MinSpan pathways better conserve the known biomolecular interactions. To further show biological relevance, we utilized the pathways alongside constraint-based modeling to experimentally discover non-intuitive novel transcription factor roles further validating the biological importance. This is only possible with our method as human derived pathways cannot be integrated with a constraint-based model and the discoveries were non-intuitive. Thus, the pathways are biologically relevant. We do however agree with Reviewers 1 and 4 that the manuscript is not as clear as desired, but we believe this to not be an issue of the actual manuscript's content but rather the presentation, which is addressable in our opinion.

We also believe that the MinSpan algorithm will become a general tool for interrogating constraint-based models, such as flux balance analysis and flux variability analysis. Many of the students in the lab are already utilizing the technique to investigate networks and many of the great and interesting points Reviewers 2 and 3 raise are in fact becoming full papers in the pipeline in our research group. Finally, Reviewer 4 previously reviewed this manuscript for Cell. They have copy-and-pasted word for word their original Cell review as they believe we have not heeded their advice. Briefly, their advice from their review at Cell was 1) the work does not have enough impact for Cell's general audience and would not rewrite textbooks, 2) they are unsure of the biological relevance, and 3) they have issues with the application of the MinSpan for experimental design. However, on nearly half of the reviewer's specific points related to parts 2 and 3 of their criticism, the manuscript that was submitted to MSB was modified from its Cell counterpart to address their concerns. In particular, we cited the Uetz et al. paper in the MSB version specifically to address their second major concern over biological importance of protein interactions and tried to clear up many of the methods for parts 2 and 3. Unfortunately, Reviewer 4 did not reread the text in enough detail to notice any of our revisions and copy-and-pasted their entire original criticism, which we believe to be an unfair review. Regarding the remainder of their criticisms, they are mainly misconceptions or factual errors which the other Reviewers did not raise and would be better addressed through a rebuttal and further clarification in the text (e.g. Reviewer 4 states in Point 3 that the computational technique we are using cannot estimate intracellular fluxes, but our group has shown with experimental evidence that our method does provide reliable estimates as already cited in the main text: Lewis et al. Nat Biotech 2010, Bordbar et al. MSB 2012, Nam et al. Science 2012). Other criticisms by Reviewer 4 are related to the impact of the work which the other reviewers all agreed that our work is an important issue for the MSB readership.

In summary, Aarash and I believe that the manuscript is well suited for the MSB readership. The pathways are biologically relevant as globally compared to the best available interaction datasets. The pathways have also been used in a prospective manner to design experiments and for biological discovery of novel transcription factor roles which were non-intuitive and only possible with the MinSpan algorithm. Finally, we believe the method will become widely used by the constraint-based modeling community. As there is a lot of excitement with new methods, there are so many great possibilities (as Reviewers 2 and 3 raise) but there needs to be an initial paper describing the work. If possible, we will gladly complete a full rewrite and rebuttal addressing all the criticisms, clearing up all misconceptions about the work, expanding the manuscript in the areas identified by the reviewers, and increasing the clarity of the manuscript.

2nd Editorial Decision

22 October 2013

Thank you again for your letter on our decision with regard to your manuscript "Minimal metabolic pathway structure is consistent with associated biomolecular interactions". We have now had the chance to read again the study, your reply letter and the reviewers comments.

We understand your point of view and acknowledge that the reviewers returned different recommendations. To summarize briefly, we appreciate that you provide in this study:

- a method to identify systematically metabolic pathways at a genome-wide scale
- a comparison of these pathways with 'classical pathways' in the context of three types of large-scale networks (PPI, GI, TRN)
- potential transcriptional new regulatory links in *E. coli*. based on computational predictions of environment-specific pathway activity.

We do agree that a rational way to systematically identify biologically relevant pathways based on a genome-scale network model is a potentially relevant topic and we appreciate the efforts in benchmarking the method with large scale datasets. However, we also agree with the reviewers' critiques that the key issue in this study is to provide compelling support for the biological and functional relevance and 'utility' of the large collection of metabolic pathways identified by such a method.

With regard to the metrics that indicate the biological functional relevance of the MinSpan pathways, we agree that the use of GI and TRN information is interesting. We appreciate that a comparison based on these metrics between MinSpan pathways and other pathway collections indicates better performance of MinSpan to recapitulate the structure of co-occurrence and co-absence of these types of interactions. We tend however to share the concerns expressed by reviewer #4 that the justification for using PPIs in the context of metabolic pathways is not immediate. We acknowledge that you cite Uetz et al 2000, but it is not very clear to us that this early large-scale Y2H study provides sufficient justification for PPI enrichment in metabolic pathways. In fact, Reviewer #1 raises again the same concern.

An important aspect missing from the study and that was requested by the reviewers (#2, #3 and #4), is a more systematical and insightful analysis of the overlap between 'classical' pathways and MinSpan pathways. How many 'classical' pathways were recovered, with what overlap? How many and which ones were missed and why?

On a related note, we understand that GI data provides, in principle, unbiased information to evaluate the purely network-based MinSpan algorithm. But if the ultimate goal is to find functional pathways, one wonders whether it would not be possible to use directly GI data to identify an optimal partitioning of the network into a set of pathways by maximizing the GI-based metric (irrespective of any consideration on the null space of S). How would such pathways that are optimized for 'functionality', according to your own metric, overlap and compare to MinSpan pathways and to 'classical' pathways?

A potentially constructive suggestion provided by reviewer #3 is to investigate whether there is an evolutionary signature for the MinSpan pathway that could provide some support to the idea that MinSpan identifies biologically functional pathways -- eg. are MinSpan pathways more conserved than 'classical' or random pathways? The types of analyses would bolster the claim made in the abstract that MinSpan identifies a "fundamental principle in the evolutionary selection of pathway structures".

We also tend to agree with the reviewers that the illustration of MinSpan pathways for the discovery of new transcriptional regulatory relationships remains somewhat preliminary. It is unclear how the three examples were chosen and whether the success rate (2/3) is reflecting the global performance of the method on a large scale. In the two positive cases (Nac and MntR), the evidence remains rather suggestive. Finally, while we find the identification of new environment-TF associations potentially interesting, it remains unclear whether it validates the novel pathways identified by MinSpan and whether this could not have been achieved with conventional pathways.

We understand your issues with the report of reviewer #4 and we apologize that this reviewer did not catch all the changes made (even though we explicitly asked this reviewer to go over the MSB manuscript again). However, we feel that several of the major issues raised by reviewer #4 as well as by the other reviewers remain relevant. Given these considerations, we feel that the biological support in favor of the method and its decisive advantages remain somewhat unclear. Resolving such issues might be possible but would involve additional analyses and experimentation with rather uncertain outcome at this stage. As such, I am afraid that we are still not convinced that the study, as it stands, would be sufficiently developed and, as such, we would not be able to revert our decision at this point.

I am very sorry not to be able to bring better news on this occasion but I hope that the remarks above explain better the reasons for our decision.

We submitted a manuscript titled: "Minimal metabolic pathway structure is consistent with associated biomolecular interactions", with tracking number MSB-13-4756 on August 5th 2013. After consideration by four referees, the manuscript was rejected and an appeal was denied.

I would like to thank you for your response to our appeal as it opened our eyes to some of the shortcomings of the work. In hindsight, we realize that the manuscript required further strengthening and more clarification and we thank you and the reviewers for your criticisms. Over the past few months, we have comprehensively completed and implemented the suggestions to improve the manuscript. As you will see in the accompanying rebuttal below, we were successful in completing all of the analyses suggested and have addressed all concerns. We ask that you reconsider the manuscript for publication.

Please see the attached document on the following pages with a full detailing of revisions from the criticisms by the editors and the reviewers.

Comments from the editor:

Thank you again for your letter on our decision with regard to your manuscript "Minimal metabolic pathway structure is consistent with associated biomolecular interactions". We have now had the chance to read again the study, your reply letter and the reviewers comments.

We understand your point of view and acknowledge that the reviewers returned different recommendations. To summarize briefly, we appreciate that you provide in this study:

- a method to identify systematically metabolic pathways at a genome-wide scale
- a comparison of these pathways with 'classical pathways' in the context of three types of large-scale networks (PPI, GI, TRN)
- potential transcriptional new regulatory links in *E. coli*. based on computational predictions of environment-specific pathway activity.

We do agree that a rational way to systematically identify biologically relevant pathways based on a genome-scale network model is a potentially relevant topic and we appreciate the efforts in benchmarking the method with large scale datasets. However, we also agree with the reviewers' critiques that the key issue in this study is to provide compelling support for the biological and functional relevance and 'utility' of the large collection of metabolic pathways identified by such a method.

With regard to the metrics that indicate the biological functional relevance of the MinSpan pathways, we agree that the use of GI and TRN information is interesting. We appreciate that a comparison based on these metrics between MinSpan pathways and other pathway collections indicates better performance of MinSpan to recapitulate the structure of co-occurrence and co-absence of these types of interactions. We tend however to share the concerns expressed by reviewer #4 that the justification for using PPIs in the context of metabolic pathways is not immediate. We acknowledge that you cite Uetz et al 2000, but it is not very clear to us that this early large-scale Y2H study provides sufficient justification for PPI enrichment in metabolic pathways. In fact, Reviewer #1 raises again the same concern.

In retrospect, we acknowledge that we had not provided adequate justification for using PPIs in this study as a metric for determining the biological relevance of pathways. Rather than point to the conjecture of literature in the previous manuscript, we now prove our point definitively through a large scale test of the conservation of PPIs in yeast metabolism. We took the known Y2H protein interaction pairs in yeast metabolism from the used data set in the text (48 total) and determined how many were in MinSpan pathways (37 pairs) and human-derived pathways (KEGG – 4, YeastCyc – 25, Gene Ontology - 47). We then generated 10,000 lists of 48 random protein pairs in yeast metabolism and repeated the analysis, finding a highly significant enrichment of true protein interactions in metabolic pathways for all four pathway types ($p < 1e-4$, empirical test). The factor of enrichment as compared to the median of the 10,000 random lists was: 3.36x for MinSpan, N/A for

KEGG (median of random lists = 0), 25x for YeastCyc, and 1.68x for Gene Ontology. This comprehensive assessment proves that protein interactions are in fact conserved in metabolic pathways.

An important aspect missing from the study and that was requested by the reviewers (#2, #3 and #4), is a more systematical and insightful analysis of the overlap between 'classical' pathways and MinSpan pathways. How many 'classical' pathways were recovered, with what overlap? How many and which ones were missed and why?

As requested, we have expanded the study to include both a global and specific analysis on the overlap and differences of 'classical' and MinSpan pathways. We speak of this broadly in terms of the global overlap and similarity of the 4 different pathway types (MinSpan, KEGG, BioCyc, and Gene Ontology). We then focus on the specific pathways missed by MinSpan or 'classical' pathways with explanations for the discrepancies (see new Figure 3, Supplementary Data, and new main text). The discrepancies are illustrated by the three key examples that we gave in the previous manuscript.

On a related note, we understand that GI data provides, in principle, unbiased information to evaluate the purely network-based MinSpan algorithm. But if the ultimate goal is to find functional pathways, one wonders whether it would not be possible to use directly GI data to identify an optimal partitioning of the network into a set of pathways by maximizing the GI-based metric (irrespective of any consideration on the null space of S). How would such pathways that are optimized for 'functionality', according to your own metric, overlap and compare to MinSpan pathways and to 'classical' pathways?

In a recent study by Dutkowski et al. 2012 (cited in main text) by Trey Ideker's group here at UC San Diego, they utilized GI, PPI, and other high-throughput data sets to infer the Cellular Component ontology of Gene Ontology, termed NeXO. The group focused on the Cellular Component, rather than Biological Processes, ontology of Gene Ontology because genetic interaction data cannot build a functional pathway, even though the Biological Processes ontology would be of greater importance (as David Botstein pointed out in the News and Views in that issue, cited in main text). This is because functional metabolic pathways have a specific order, from substrate to product across multiple metabolic reactions that cannot be captured by GI data.

In addition, Gene Ontology already contains pathways based on many data resources, including physical interactions (PPIs) and genetic interactions. As we have shown, MinSpan globally outperforms such pathway or gene classification databases.

A potentially constructive suggestion provided by reviewer #3 is to investigate whether there is an evolutionary signature for the MinSpan pathway that could provide some support to the idea that MinSpan identifies biologically functional pathways -- eg. are MinSpan pathways more conserved than 'classical' or random pathways? The types of analyses would bolster the claim made in the abstract that MinSpan identifies a "fundamental principle in the evolutionary selection of pathway structures".

We have determined the conservation of the MinSpan pathways vs. random pathways, random matrices, and 'classical' pathways across *E. coli* and *S. cerevisiae*. The results are briefly mentioned in the Discussion and our analysis is presented in full in the supplementary material.

In brief, MinSpan pathways are well conserved, as compared to random MinSpan pathways (RandSpan) (345% enrichment) and to random matrices with the same distribution of non-zero entries as MinSpan pathways (83% enrichment) (Figure S5C). The pathways are marginally less conserved as BioCyc and Gene Ontology (28% and 20% respectively). The higher conservation of human-defined pathways are due to: 1) human defined pathways are typically universally chosen, and 2) human-defined pathways are built based on topology of conserved enzymes and not metabolic functionality. MinSpan pathways take into account both network topology and metabolic flow (function). Isotopomer flux profiling studies have previously shown that the combination of genome-annotation (topology) and flux profiling (functionality) is needed to define metabolic pathway functionality as the function of a pathway may not be intuitive from the topology alone (Amador-Noquez et al., 2010).

We also tend to agree with the reviewers that the illustration of MinSpan pathways for the discovery of new transcriptional regulatory relationships remains somewhat preliminary. It is unclear how the three examples were chosen and whether the success rate (2/3) is reflecting the global performance of the method on a large scale. In the two positive cases (Nac and MntR), the evidence remains rather suggestive. Finally, while we find the identification of new environment-TF associations potentially interesting, it remains unclear whether it validates the novel pathways identified by MinSpan and whether this could not have been achieved with conventional pathways.

We apologize for not being clear on this point. This workflow is only possible with MinSpan and not conventional pathways as MinSpan can directly couple flux distributions calculated by constraint-based models (FBA, FVA, sampling, etc.) with pathway properties. We have updated the text to show this point:

“MinSpan is an inherent property of metabolic networks, unlike human-defined pathways, and offers the direct ability to predict pathway associated biomolecular properties from flux distributions calculated by constraint-based modeling.” (Lines 317-319).

“The analysis presented to predict TF-environment associations is only possible with MinSpan pathways, as opposed to pathway databases, as MinSpan pathways directly link flux simulations from constraint-based models to pathway biomolecular properties.” (Lines 433-435).

Further, the three experiments were specifically chosen to provide the toughest test for MinSpan. We chose novel predictions in very well studied environmental shifts to make it extra difficult for us to discover new transcriptional regulation. The text has been updated as below:

“We chose three novel TF-environment predictions to experimentally validate that are non-obvious, in the sense that little to no literature links the TF with the predicted associated environment. To be rigorous in the experimental design, we chose environmental shifts that have been well-studied; where discovering novel experimental findings would be more difficult.” (Lines 371-375)

Finally, yes, we agree that the results are only 3 of the dozens of potential experiments. However, we feel that the enabler of the experiments is MinSpan, a computational algorithm. Completing such dozens of experiments would be its own manuscript, and if 1000s of new non-intuitive regulatory events were discovered would be quite a substantial paper on its own.

We understand your issues with the report of reviewer #4 and we apologize that this reviewer did not catch all the changes made (even though we explicitly asked this reviewer to go over the MSB manuscript again). However, we feel that several of the major issues raised by reviewer #4 as well as by the other reviewers remain relevant. Given these considerations, we feel that the biological support in favor of the method and its decisive advantages remain somewhat unclear. Resolving such issues might be possible but would involve additional analyses and experimentation with rather uncertain outcome at this stage. As such, I am afraid that we are still not convinced that the study, as it stands, would be sufficiently developed and, as such, we would not be able to revert our decision at this point.

All the remaining issues are rebutted in great detail or thoroughly revised in the manuscript. With respect to reviewer #4, we believe most of their comments are either 1) factually incorrect, 2) due to misunderstandings about the work, or 3) changes that they missed to see in the MSB version of the text.

Comments from reviewers:

We thank Reviewer #1 for their constructive criticism. We have implemented their suggestions.

Reviewer #1 (Report):

This paper addresses an important issue, namely how to define biologically meaningful metabolic pathways in an unbiased manner. The MinSpan pathways potentially fulfil this criterium. I have some major concerns however. As it stands it is difficult to grasp the essence of the MinSpan pathways and I miss a large part of the biological support which the authors claim to have. This is at least partly due to a superficial and inaccurate way of formulating and partly because of a lack of data and/or unclear presentation of the data.

Specific comments

In the introduction the authors address the limited applicability of convex-based pathway definitions, such as EFMs, due to combinatorial explosion. This combinatorial explosion has been analysed and brought back to multiple pathways in just a small set of subnetworks (DOI: 10.1038/srep00580). Could the authors comment on this work and relate it to the MinSpan pathways?

We apologize for not citing this work. At Lines 82-84, we had cited key papers that had modified elementary flux modes (EFMs) to deal with the combinatorial explosion. The papers we had cited were elementary flux patterns (EFPs, which subset by metabolic subsystem, Kaleta et al. 2009) and k-shortest pathways (which calculates a user defined number of pathways from the total possible by convex analysis, de Figueredo et al. 2009). We forgot to cite the work by Kelk et al. (CoPE-FBA) as Reviewer 1 states. We have added this citation at Lines 82-84 with the other two citations.

However, we disagree that this paper deals with the combinatorial explosion of convex-based pathway definitions. Briefly, in the work by Kelk et al., the authors constrain the objective function of the model to its optimal value, and then calculate convex pathways using a rendition of EFMs. This work focuses on the sub-region of the optimal solution space. Similar to the two other works that we cite, this focuses only on a subset of the metabolic pathways defined by the null space and not the totality of phenotypic states achievable by the model. If the objective function flux is changed, even by an arbitrarily small amount, the calculated pathways are no longer applicable. In addition, the Kelk et al. text focuses most of their work using the much older and smaller *E. coli* model, iJR904. In Table 1 of their text, when they expand to iAF1260 (similar in size to the newer iJO1366 that we study), they calculate 1,679,616 pathways with only glucose as the input. These pathways only describe the optimal solution space and not any other phenotype.

On the other hand, MinSpan pathways capture the entire solution space for all metabolite inputs and outputs at once. This entire space is described with only 750 MinSpan pathways. This can be seen as a comprehensive and concrete set of pathway definitions that definitively resolve what was previously an intractable combinatorial explosion. The work of Kelk et al. is useful but ultimately subsets the problem similar to EFPs and k-shortest pathways and is unable to address the key aspects of the combinatorial explosion.

The MinSpan algorithm is exactly described for computational systems biologists, but an insightful interpretation for a wider systems biology audience is lacking. The small example gives limited insight due to the oversimplification of the pathway (Fig 1). All reactions are unimolecular, while real metabolic networks are highly connected via cofactors such as ATP or NAD(P)H. The simplified TCA cycle cannot carry any flux at steady state. Unlike the real TCA cycle it has no net input or output and can therefore only evolve to equilibrium. It is unclear why the same reaction is sometimes represented as irreversible and sometimes as reversible in the decomposition shown in Fig 1. The comparison to the EFMs is highly relevant and should better be included in the main text rather than in the supplement (Fig S1 and S2). The explanatory power of this paper can be greatly improved by the use of a more realistic example, including interconnections via a cofactor, and by a more exact definition of the MinSpan pathways.

The toy model that we used is of historical importance. It has been showcased in various publications, including Papin et al. 2003 *Trends in Biochem Sci.* and Systems Biology by Palsson.

However, we agree with the reviewer that this toy model is an oversimplification. We have updated the toy model to include cofactors (ATP, ADP, NADH, and NAD) and have rerun the analyses and updated the figures and text as requested.

Though we agree that the comparison to convex optimization techniques (e.g. EFMs and ExPas) is very important, we think that figure should remain in the Supplementary Material due to space constraints. Adding 16 additional pathways to Figure 1 would add a lot of clutter and further confuse readers new to pathway analysis. We have updated the manuscript at (Lines 122-124) to alert the reader that the comparison with ExPas and EFMs is available in the Supplementary Material.

With respect to Figure S2 above: From the legend I have no clue how this represents a comparison between MinSpan and extreme pathways. Please, add a descriptive legend explaining what is exactly presented.

We apologize for the lack of clarity of the Supplementary Figure Legend for Figure S2. A more detailed understanding was presented in Section 2 of the supplement but was not carried over into the legend. We have gone ahead and changed this. The new legend reads as follows:

Figure S2: Comparison of MinSpan pathways and Extreme Pathways for the *E. coli* core mode are presented. (A) The 23 MinSpan pathways for this model were hierarchically clustered and placed into subsystem categories of glycolysis, anaplerotic pathways, fermentation, TCA, and pentose phosphate pathway. The MinSpan pathways are on average 20.8 reactions in length. (B) There are 16690 Extreme Pathways which were hierarchically clustered into 50 groups. The number of pathways per group and average length of pathways within that group are shown. The MinSpan pathways are a subset of the Extreme Pathways. The coloring on the dendrogram branch relates the location of the MinSpan pathways in the Extreme Pathway groups.

Biological support comes from different kinds of interaction networks. I miss an exact explanation as to why these networks support the MinSpan pathway definition. For TF networks this is rather intuitive, but protein-protein interactions are surprising. Metabolites diffuse from one enzyme to the next and channeling has been demonstrated only in rare cases. Hence a physical interaction, as measured by the 2-hybrids, is not essential for proper functioning of a pathway.

We apologize for not providing a more definitive justification for the use of the different kinds of interaction networks in determining biological relevance of the MinSpan and human-derived pathways. We are not trying to argue that a metabolic pathway must contain a biomolecular interaction (protein, genetic, or transcriptional). Instead, we are stating that such types of interactions should be enriched in a metabolic pathway system. Transcriptional regulation is conserved in pathways (as described by numerous papers including Wessely et al. MSB 2010), as well as positive genetic interactions (both by definition and numerous papers including Kelley and Ideker Nat Biotech 2005). However, there is contention from Reviewers 1 and 4 that protein interactions would be conserved in pathways. We apologize for not providing a more robust justification for the use of protein interactions in our correlation analysis.

To test whether protein interactions are conserved in metabolic pathways, we took the known Y2H protein interaction pairs in yeast metabolism (48 total) and determined how many were in MinSpan pathways (37 pairs) and human-derived pathways (KEGG – 4, YeastCyc – 25, Gene Ontology - 47). We then generated 10,000 lists of 48 random protein pairs in yeast metabolism and repeated the analysis, finding a highly significant enrichment of true protein interactions in metabolic pathways for all four pathway types ($p < 1e-4$, empirical test). The factor of enrichment as compared to the median of the 10,000 random lists was: 3.36x for MinSpan, N/A for KEGG (median of random lists = 0), 25x for YeastCyc, and 1.68x for Gene Ontology. This comprehensive assessment proves that protein interactions are in fact conserved in metabolic pathways.

We have updated the main text in the manuscript (Lines 127-138) to further clarify and justify the correlation analysis against biomolecular interactions, particularly protein interactions. We have also added a new supplementary material section discussing the analysis for conservation of Y2H protein interactions, a few examples of Y2H protein interactions found in metabolic pathways, and Figure S3.

The flux calculations with the Cobra toolbox are hardly described. Were any modifications, even small updates, made to the cited models? If so, SBML files should be included. Which 52 nutrient conditions were applied and which constraints were imposed? Are there any particular settings in the algorithm that influence the outcome? Please, give an explanation that enables a qualified scientist to repeat the work.

These are all important points that would aid in the repeatability of this work, and we appreciate the reviewer carefully vetting our work to identify the details that were not adequately specified. We have updated the methods to detail these items. Briefly, we note that no modifications were made to the *E. coli* model (iJO1366) for sampling. To limit any confusion for the particular exchange constraints used, we have added a new supplementary excel file (sampleConditions.xlsx) that has the exchange conditions for all 52 nutrient conditions.

With respect to figure 4 where calculated TF activity is correlated to calculated flux, it is claimed that 37 out of 51 nutrient shifts matched known associations (true positives). What is the exact criterium for a true positive, i.e. what is considered minimum information to call an association a true positive? I have been looking for a list of associations and the proof for them being true/false or undetermined, but I did not find it. In the supplement I only find a general list of papers cited, but no quantitative analysis of the results.

The reviewer raises an important concern, and we have now have clarified this point. Specifically, we have updated the Supplementary Material with 1) the criteria for a good/bad/marginal prediction, 2) the full list of predictions on all 51 nutrient shifts, and 3) the EcoCyc/literature true positives and false negatives. The criteria and explanation is in the Supplementary Material pdf and the full list of predictions and EcoCyc matches are in a Supplementary Data File (tfShiftResults.xlsx).

The authors report that global analysis of the gene sets suggested that the predictions for ade/Nac and O2/MntR were correct, but I cannot evaluate the validity of this statements. To this end RNAseq datasets were generated. However, in the Materials and Methods only limited information is given about the experimental setup and the results are not shown.

We have now made several edits to the manuscript to improve clarity on all of these points. First, we had a GEO accession set up but had forgotten to include it in the manuscript acknowledgments. It is now there and can be accessed:

<http://www.ncbi.nlm.nih.gov/geo/query/acc.cgi?token=nlqvtgsmuogmxm&acc=GSE48324>

Second, new text has been added in Materials and Methods (Lines 665-668). For each differential gene set, we determined if the genes were enriched for any particular transcription factor. A prediction was called correct if the enriched transcription factors are already known to be involved in that nutrient shift.

As is already in the text, in the ade/Nac case, GcvA, Lrp, and PurR were enriched. PurR is the main regulator of purine metabolism. GcvA is also involved in the regulation of purine metabolism. Looking at the drill down analysis, we saw that Lrp was in fact being regulated by a small rRNA (gcvB).

For L-trp/Cra, no transcription factors were enriched which we deemed as an inconclusive prediction. Finally, for the O2/MntR case, transcription factors known to be associated with anaerobic/aerobic shift (e.g. ArcA and Fnr) were enriched.

Third, we have added a Supplementary File (differentialGeneSets.xlsx) which contains the list of the gene sets for each of the three experiments.

Throughout the paper, I often wonder what a sentence means exactly. The paper could be improved substantially by checking the clarity of each statement. This also holds for figure legends, which give conclusions but often lack sufficient information to understand what is precisely depicted in the

figure.

We have carefully revised the text to improve clarity.

We thank Reviewer #2 for their constructive criticism. We have implemented their suggestions.

Reviewer #2 (Report):

In their work, Bordbar et al. propose a new definition of metabolic pathways which are derived computationally in a maximum parsimony framework from a genome-scale metabolic network. This addresses an important problem in metabolic research, namely the decomposition of a metabolic network in minimally functioning units. While traditionally classically defined metabolic pathways have been used for this purpose, these type of metabolic pathways often hamper new discoveries e.g. if pathways usually not considered together in the classical context are interacting. While approaches such as elementary flux modes and extreme pathways have been proposed to overcome these limitations, they are not applicable to genome-scale metabolic networks. In this context, the approach presented by the authors represents an important step for a novel automated definition of metabolic pathways. They demonstrate that their pathways are more representative than classically defined pathways based on the analysis of interactions in several large-scale datasets. However, this problem might be remedied by alternative approaches such as partially coupled reaction sets. There are only a couple of issues I would like the authors to address relating to the uniqueness of the MinSpan pathways and the representation of metabolic pathways.

Major issues:

It is not entirely clear to me how "stable" the MinSpan pathways are. Since they are based on a MILP-formulation, alternative solutions might exist. Moreover, small modifications in the stoichiometric matrix such as adding or leaving out a reaction might give rise to entirely different MinSpan pathways. Thus, the authors should discuss whether they will always find very similar pathways even if they "perturb" the metabolic network. Also they should analyse whether alternative optima of the MILP-approach exist and to which extent these optima differ from each other.

MinSpan pathways are dependent on the metabolic reconstruction used. We have added a paragraph in Discussion on this limitation. Reviewer #2 is correct that an MILP approach could have alternative optimal solutions. The number of non-zero entries in the MinSpan pathway matrix is unique, but the exact pathway structure can change.

We have now included additional analysis, and provided text to discuss the issue of alternate optima in the Materials and Methods (Lines 552-556) and refer the reader to the supplementary material. We have developed an algorithm to enumerate alternate optima. We generated 1000 alternate MinSpan pathway matrices for both *S. cerevisiae* and *E. coli*. The 2000 pathway matrices vary little between the two original calculated MinSpan matrices (on average, 0.66% different reactions per pathway in *S. cerevisiae*, and 0.26% different reactions per pathway in *E. coli*). With the majority of pathways being 20-30 reactions in length in both models, for every ten pathways, one reaction is different. Further, we re-ran the correlation analysis used for assessing biological relevance. There is very little difference between the results across the alternate optima and the overall biological relevance results do not change depending on which of the pathway matrices is used.

It is difficult to assess in a non-arbitrary fashion whether small modifications to the stoichiometric matrix will affect the MinSpan pathway results. As Figure S5A and S5B show, most reactions are used in only in a couple pathways. If these reactions were removed, only one or a few MinSpan would be lost or changed. However, some reactions are used in over 100 pathways. If one of those reactions was removed, then not only would the MinSpan pathways considerably change, but the entire model would change as many reactions would completely lose flux.

Also along these lines I would be interested in "false negatives", that is, to which extent classically defined pathways (e.g. in EcoCyc) are not reflected by the MinSpan pathways.

We have added a new section in the text globally comparing the similarities and differences of MinSpan pathways with classical pathways. The “false negatives” are highlighted in that new section (Lines 209-267 and Figure 3). The different pathways are also available in the Supplementary Files (pathwayDifferences.xlsx).

The approaches presented by the authors should be made available, e.g. as component of the COBRA toolbox.

As described in the Methods of the algorithm (Lines ~556), “MinSpan is available for COBRAPy at (<https://github.com/sbrg/minspan>).” The code is currently not available at this URL but will be made available as soon as the paper is accepted.

Minor issues:

There appears to be an error in the RNAseq group assignment: there are two groups (E2 and G2) with the same conditions.

Thanks for pointing out this mistake. We have updated the text accordingly, G2 was incorrect and should have read as (WT+nutrient and KO+nutrient).

Though the shortest path between two major metabolites can often be the most accurate representation, it does not have to be (p. 18)

We have removed this sentence completely.

No explanation of "type 3 loop" (p. 23)

Thanks, we have updated this sentence to give a general idea of type 3 loops (Lines 618-619) and a citation to Systems Biology by Palsson in 2006.

A accession number for the RNAseq data should be provided.

We apologize for not including the URL in the first submission. The data was already available for reviewers at GEO but we forgot to include the website:

<http://www.ncbi.nlm.nih.gov/geo/query/acc.cgi?token=nlqvtgsmuoogmxm&acc=GSE48324>

We thank Reviewer #3 for their constructive criticism. We have implemented their suggestions.

Reviewer #3 (Report):

Bordbar et al. present an interesting new approach for the unbiased definition of metabolic pathways. The new method scales linearly with the size of metabolic networks, allowing for a structural definition of pathways that can sustain fluxes at steady-state on a genome scale. The predicted pathways correlate well with data on protein and gene interaction networks, and predict potential gene-transcription factor associations. I think this is an interesting contribution and probably deserves publication.

Main comments:

1. Although the authors provide a description of global pathway properties in figure S3, I feel it is important to provide this or similar description in the main text. In particular, the typical pathway sizes and the number of pathways per reaction should be discussed. The method enumerates a large number of pathways in both organisms tested, but one immediate question is whether these pathways have some clear biological characteristics or if there are many spurious pathways that would be hard to interpret.

We have expanded the text to include a new section comparing the number and size of pathways in all of the databases (Table 1 and Figure 3). This section also discusses what pathways MinSpan captures and misses, and the differences between all the databases. Figure S3 (now Figure S5) remains in the supplement as a new part of a supplemental section related to the homology of pathway structure between *E. coli* and *S. cerevisiae*.

2. *While the authors point out that historically biochemical pathways have been defined in a universal fashion rather than being organism specific, there is no clear reference as to how minSpan pathways discovered in two organisms compare to each other. One reason classical biological pathways are popular is that they are generally consistent between species. Are minSpan pathways well or poorly conserved in multiple species?*

This is a good idea and a graduate student in the lab is already working on a project comparing MinSpan pathways across dozens of close and distant species (based on phylogeny). As a preliminary analysis for this manuscript, we have determined the conservation of the MinSpan pathways between *E. coli* and *S. cerevisiae* as they are the pathways used in the main text. MinSpan pathways are much better conserved than random pathways (RandSpan) and random matrices with similar distributions of non-zero entries (345% and 83% enriched, $p = 3.91e-220$ and $p = 2.31e-101$, Kolmogorov-Smirnov test). We also compared MinSpan to BioCyc and Gene Ontology (KEGG was omitted from this analysis as KEGG does not contain organism-specific pathways). BioCyc and Gene Ontology pathways are marginally better conserved (28% and 20% enriched, $p = 1.07e-7$ and $p = 3.74e-6$ respectively). The pathways are not as conserved as BioCyc and Gene Ontology. The higher conservation of human-defined pathways are due to: 1) human defined pathways are typically universally chosen, and 2) human-defined pathways are built based on topology and not metabolic functionality.

A strength of human-defined pathways can be their universality, especially for educational purposes. However, universality can also be a weakness as focusing on topology ignores systemic functionality. Isotopomer flux profiling studies have shown that the combination of genome-annotation (topology) and flux profiling (functionality) is needed to define true metabolic pathway functionality as function can be different than the specific gene products (Amador-Noquez et al., 2010). Further, as the main text results show, the MinSpan algorithm is better at conveying biomolecular data, suggesting that a universal approach to defining pathways is not accurate, as metabolism varies between organisms.

The limitations and analysis of conservation are presented in the final paragraph of the Discussion (Lines 494-504) and in the Supplementary Material.

3. *The authors attribute the larger number of pathways in E. coli to a more complete metabolic network, implying that the definition of minSpan pathways depends on the level of curation and pre-existence of genome-scale metabolic models. How sensitive are minSpan pathways to the level of curation in different models? Even for the same organism, different reconstructions are published by different groups, and different models are of varying quality. These limitations should be discussed.*

We have added a new paragraph at the end of the Discussion with limitations of the approach (Lines 485-494). The main limitation as noted by Reviewer #3 has been mentioned, particularly that the MinSpan algorithm is dependent on the metabolic model used. Better (or worse) pathways will be calculated based on the quality of the reconstruction. This of course parallels how traditional pathways are defined as well, as they are dependent on the amount of biochemical knowledge of the particular organism, which is also reflected in the quality of most reconstructions.

Reviewer #4 (Report):

I have already reviewed this manuscript when it was submitted to a different journal. As major concerns remain unaddressed, I copy-paste here my last recommendation:

Our existing biochemical pathways for metabolism are human-made and they are thus not necessarily a representation of the actual underlying biological function. Here, (i) the authors develop a computational algorithm, with which they can extract functional units/pathways from genome-scale metabolic networks, (ii) they argue that these extracted MinSpan pathway correlate better with data on protein-protein interactions, transcriptional networks, and genetic interactions than human-made pathways do and (iii) they use the new pathways to identify few new transcription factor interaction

sites. While I see some value in their work, I feel that this manuscript is more suitable for a specialized journal.

Furthermore, I have concerns about the general validity of their claims.

1. Importance? Relevance?

"Pathways" are required for biochemical education and, for instance, for the analysis of experimental (omics-) data (to provide "context" for the analysis). While the new MinSpan pathway definition seems to have some advantages over the human-defined pathways (e.g. as shown in Figure 3A, B), a number of questions arose, which make me doubt that the new pathway concept will have a widespread impact:

a) It is not clear how "reasonable" the pathways are that the authors find. They mention some "reasonable" ones, but what about the rest? What is the degree of agreement between the human-defined ones and the new ones? Do we get back the TCA cycle, glycolysis, and pentose phosphate pathway?

We have completed a rigorous comparison of MinSpan with traditional pathways (Lines 209-267 and Figure 3). MinSpan captures most traditional pathways and in the case of *E. coli* add new pathways (Figure 3D). There are some representative differences which are due to reasons previously explained through the three examples, which are retained in the text. We have also added a Supplementary Data file containing all pathways missed by MinSpan and the reasons why.

b) In any case, I have very strong doubts that our biochemical textbooks are going to be rewritten; further I doubt that the widely used resources such as KEGG, BioCyc, etc. are going to replace the human-made pathways by the MinSpan pathways.

c) If this doesn't happen, then the only point that is left from the authors' work is a tool to computationally define new pathways that can then be used to analyze, e.g. omics, data, where we so far use human-defined pathways. However, it is not clear whether the new pathways are so much more powerful as compared to the old ones.

Overall, while I see some value in the authors work (because the new pathway concept could make one or the other computational algorithm a bit better), I feel that the work is not groundbreaking enough to be published in MSB. I feel that the work is more appropriate for a specialized journal.

These remarks are left over from the reviewer's comments for Cell. The reviewer believes that we did not take into consideration their advice in previous reviews. However, the reason we did not contest the decision at Cell is that we fully agreed with the reviewer that this work was not as of broad interest as we initially thought.

However, we maintain the belief that the work will be of interest to the readership of MSB. In addition, the three other reviewers agree with us that this work is in fact of interest to MSB readers.

2. Claims fully supported?

The authors claim that their MinSpan pathways reflect the biological reality better than the human-defined pathways. While this is a point that most people would generally be tempted to believe, I am not sure whether the authors present sufficient evidence that their claim is actually true:

a) Their work rests on the assumption that a functional pathway needs to correlate with PPI, genetic interactions, and co-regulation. However, they do not present any evidence for this assumption. E.g. they state "As most PPIs occur between proteins within the same metabolic complex or adjacent metabolic reactions ...", but they don't provide a reference for this statement.

We apologize for not providing a more clear explanation for the use of the different kinds of interaction networks in determining biological relevance of the MinSpan and human-derived pathways. We are not trying to argue that a metabolic pathway must contain a biomolecular interaction (protein, genetic, or transcriptional). Instead, we are stating that such types of interactions should be enriched in a metabolic pathway system. Transcriptional regulation is conserved in pathways (as described by numerous papers including Wessely et al. MSB 2010), as well as positive genetic interactions (both by definition and numerous papers including Kelley and Ideker Nat Biotech 2005). However, there is contention from Reviewers 1 and 4 that protein interactions would be conserved. We apologize for not providing a more robust justification for the use of protein interactions in our correlation analysis.

To test whether protein interactions are conserved in metabolic pathways, we took the known Y2H protein interaction pairs in yeast metabolism (48 total) and determined how many were in MinSpan pathways (37 pairs) and human-derived pathways (KEGG – 4, YeastCyc – 25, Gene Ontology - 47). We then generated 10,000 lists of 48 random protein pairs in yeast metabolism and repeated the analysis, finding a highly significant enrichment of true protein interactions in metabolic pathways for all four pathway types ($p < 1e-4$, empirical test). The factor of enrichment as compared to the median of the 10,000 random lists was: 3.36x for MinSpan, N/A for KEGG (median of random lists = 0), 25x for YeastCyc, and 1.68x for Gene Ontology. This comprehensive assessment proves that protein interactions are in fact conserved in metabolic pathways.

For the manuscript, we have updated the main text (Lines 127-138) to further clarify and justify the correlation analysis against biomolecular interactions, particularly protein interactions. We have also added a new supplementary material section discussing the analysis for conservation of Y2H protein interactions, a few examples of Y2H protein interactions found in metabolic pathways, and Figure S3.

b) Even if this assumption is true, I am not sure whether there is sufficient statistical evidence that the MinSpan pathways are better, although this is crucial for their whole story. First, unfortunately, on the basis of the provided information, it is impossible to understand what the authors did to come to the key figures 2D-E. E.g. it is unclear what the y-axis in Figure 2D, E and F mean. The applied statistics are described with only three cryptic sentences on page 23, which are insufficient to allow for solid evaluation of the validity of the claim; e.g. what kind of "filter" was applied? What is the "total number of correlations"?, what is a threshold parameter? Etc. Furthermore, I do not understand that the "random" area is so small and why it sits where it sits.

A Receiver Operating Characteristics (ROC) curve is a classical statistics tool used in bioinformatics and systems biology. They have been used in many publications including some in Molecular Systems Biology (e.g. Jerby et al. 2010) and are applicable in this scenario. We do not feel there is anything wrong with our statistical analysis or its presentation.

As to their specific comments:

In the section, “Correlation Analysis” in Materials and Methods, the reviewer’s concerns are clearly explained. Their concerns are:

1) *“e.g. what kind of filter was applied”*

From the Text: Pairwise Spearman rank correlations of the co-occurrence or co-absence of genes and proteins across the pathways were calculated. Correlations that were not significant ($p > 0.05$, permutation test) were filtered from further analysis (Lines 567-568).

2) *“what is the total number of correlations”*

From the Text: The total number of correlations remaining after filtering is the Coverage criteria in the x-axes of Figure 2. (Lines 571-573)

3) *“What is a threshold parameter?”*

From the text: The correlation coefficient was the varied discrimination threshold for generating the receiver operating characteristics (ROC) curve (Lines 573-574).

4) *I do not understand that the "random" area is so small and why it sits where it sits.*

“Random” is RandSpan as it is clearly labeled on Figure 2 and is clearly described in the main text.

Second, the more illustrative (non-statistical) support that they use to back up their claim (i.e. Figures 3A-C) is nice and helpful, but it is completely unclear whether the shown examples are just a few hand-picked anecdotal examples, which by accident made sense, or whether there would be many, many more of such findings.

The global statistics from the correlation analysis definitively show that MinSpan pathways have more biological support than classical pathways and that we are not cherry-picking our results. We decided to provide illustrative examples to help the lay reader understand the differences.

Examples by definition are hand-picked. However, as we have added a larger, global comparison of MinSpan and classical pathways, we have classified the differences in pathways using the criteria described in these three examples.

3. Problem with the second part:

In the second part, the authors make prediction in reaction fluxes upon environmental/genetic perturbations, then they use MinSpan pathways to provide context for the flux changes and search for enrichment of certain transcription factors in the changed MinSpan pathways. Generally, I have a number of problems with this second part:

a) The integration with the rest of the text is not optimal; i.e. the story drifts away from MinSpan. In fact, this part reads as if it was written by a different person...

*b) In this second part, the authors need to computationally estimate intracellular fluxes in a number of mutants and under a number of different environmental predictions: Reasonable predictions of changes in reaction fluxes can (assuming that the applied sampling approach can in fact yield reasonable results) only be done if the respective substrate uptake rates are known. First, it is not clear what the authors have precisely done in their simulations (because the text is a bit cryptic, e.g. "For alternate carbon ... the original source was set to zero and an equal amount of atoms was set as the input? Do they mean "amount" or do they mean "rate"? Do they mean *carbon* atoms? What do they mean by "fluxes were normalized by median growth rate"? Where did they get the growth rates from?). Second, whatever they have done, growth rates cannot be a priori predicted by FBA (unless one uses experimentally determined uptake rates; or, if one is interested in flux changes between conditions, if one knows the changed uptake rates between conditions). As I do not see that the authors have done this, it is not clear how they can reliably estimate difference in intracellular fluxes, which is a necessary requirement to determine the flux load of the MinSpan pathways, which in turn was necessary for their statistical analysis shown in Figure 4B.*

It has been shown in multiple studies (Nam et al. Science 2012, Bordbar et al. MSB 2012, Lewis et al. Nat Biotech 2010) that substrate uptake rates are not needed to accurately estimate relative changes in reaction fluxes. Substrate uptake rates are only needed for absolute flux levels. These studies were all cited in the original version of the text (Lines 327-328). In all three studies, the estimated relative fluxes based on Markov Chain Monte Carlo sampling were rigorously compared to omic data sets (transcriptomics, proteomics, and metabolomics) to confirm the accuracy of the predictions.

We thank the reviewer for pointing out that some of our methods were unclearly described. We had changed these prior to the initial submission to MSB and many of these comments are left over from their previous review to Cell.

c) Generally, this part of the manuscript is not fully clear; for example:

a. It is not fully clear how the results shown in Figure 4B were generated (pages 14/15).

We apologize for not including the exact metric. The transcription factor activity for Figure 5B is calculated based on the percentage of differential MinSpan for a particular shift that are associated with that TF. This has been clarified in the legend for Figure 5 (Lines 948-949).

b. The transcription factor MntR is - according to Ecocyc - associated with 4 proteins (<http://www.ecocyc.org/ECOLI/NEW-IMAGE?type=ENZYME&object=CPLX0-7672>). However, none of these 4 proteins have anything to do with the metabolic changes that I would expect to happen with a shift to anaerobic conditions - which is where the authors would predict that this TF does something. Thus, I wonder how it happened that this TF was found to be connected with MinSpan pathways that would need to change when shifting from aerobic to anaerobic conditions? But eventually, I have missed something

The whole idea of this portion of the text is to test a transcription factor (e.g. MntR) in a condition that has nothing to do with its current literature (e.g. the anaerobic shift). MntR primarily regulates manganese transport through mntH. But mntH is also an iron transporter. Both manganese and iron

are important ions in electron transfer which is affected during oxygen shifts, whose computed fluxes are affected during aerobic to anaerobic simulations.

c. Further, it is not clear how the authors predicted the new TF-gene interactions mentioned on page 17.

d) Generally, I guess the authors would like to use this part as a demonstration that the use of their new pathways can support new biological discovery. However, here they followed up on three randomly picked (?)TF-environment interactions and on the basis of these, it is not clear whether they have just been lucky to find new biology.

We apologize for not being clear enough in our text. The purpose of the experimental portion of this study was to rigorously test non-intuitive MinSpan predictions to find novel regulation. We have rewritten the text as follows:

“We chose three novel TF-environment predictions to experimentally validate that are non-obvious, in the sense that little to no literature links the TF with the predicted associated environment. To be rigorous in the experimental design, we chose environmental shifts that have been well-studied; where discovering novel experimental findings would be more difficult.” (Lines 371-375)

These are not three randomly picked interactions. These experiments were picked specifically to give us the least chance of success to rigorously test our method.

As previously mentioned in the text, we also chose these three TF-environments because they are very different from each other, in terms of the magnitude of the perturbation caused by the knockout or the environmental shift.

Furthermore, it is not clear whether the same discoveries would not also have been possible with human-made pathways.

We apologize for not being clear on this crucial point. The driver of the experimental portion of the text is the calculation of relative flux changes using constraint-based modeling during environmental change. The MinSpan pathways are directly linked to the flux distributions as the MinSpan is a representation of the model’s null space. Human-defined pathways are not linked to calculated fluxes, and hence cannot directly interpret the relative flux changes.

We thank the reviewer for pointing out that this was not clearly described. We had addressed this comment prior to the initial submission to MSB (Lines ~420, “The analysis presented to predict TF-environment associations is only possible with MinSpan pathways, as opposed to pathway databases, as they are directly linked to constraint-based metabolic models.”) This criticism is left over from their previous review to Cell.

However, we decided to further emphasize this point in the new draft:

“MinSpan is an inherent property of metabolic networks, unlike human-defined pathways, and offers the direct ability to predict pathway associated biomolecular properties from flux distributions calculated by constraint-based modeling.” (Lines 317-319).

And updating the previous revision that was sent to MSB.

“The analysis presented to predict TF-environment associations is only possible with MinSpan pathways, as opposed to pathway databases, as MinSpan pathways directly link flux simulations from constraint-based models to pathway biomolecular properties.” (Lines 433-435).

Minor comments:

1. How many of such MinSpan pathways exist? How does it look in central metabolism? In linear biosynthesis pathways?

As described above, a new section has been added that covers the number of pathways and the overlap with traditional databases. (Figure 3, Table 1, and main text)

2. *The authors argue that a functional unit of a metabolic pathway should have protein interactions. What is the basis for this assumption?*

This comment is very similar to a major comment raised above and has been addressed above.

3. *Why are the data points of "Biocyc" and "KEGG" in the Figures 2D-F so much different, while the networks in these databases are predominantly the same human-derived ones?*

These two pathway databases are quite different. We have updated the text comparing all the pathway systems (Table 1 and Figure 3). Differences in these databases occur because human biases adding during pathway definition and whether the pathway database is organism specific or universal.

4. *Not 100% clear what the nutrient shifts are? Is this always glucose + something? Or only the mentioned nutrients?*

The nutrients shifts are clearly delineated in the Materials and Methods section. We have also added a Supplementary File (samplingConditions.xlsx) that contains the exact media composition for each simulation.

5. *On page 15, the authors say that 37 out of the 51 nutrient shifts matched known associations ... It is not 100% clear how they got to this statement?*

We have added a Supplementary Section and a Supplementary File (tfShiftResults.xlsx) that clearly shows the criteria and the results for obtaining the results.

6. *While in the text, the transcription factor Cra is called Cra, in Figure 4B the authors use the old term FruR. This is extremely confusing for readers who are not so familiar with E. coli transcriptional regulation.*

This comment was addressed prior to the initial MSB submission and is left over from their previous review to Cell.

7. *In Figure 4B, I would suggest that the authors should highlight the discussed interactions such that the reader doesn't have to search them.*

We have updated the figure by placing a box around the three TF-environment associations that were experimentally tested.

8. *Last sentence on page 16: "all three TF": At first, it is not clear, to which three TF are the authors referring to.*

We apologize that we were not clear enough. We have changed the phrasing to: "all three tested TFs". This now clearly pinpoints the three TF/environmental conditions tested, which is the focus of that particular section.

9. *Are "receiving operating characteristics" not rather called "receiver operating characteristics"?*

This comment was addressed prior to the initial MSB submission and is left over from their previous review to Cell.

10. *Referencing to the supplementary information is badly done in the text. In fact, the supplementary information reads partly like left over text. For example, I do not know what the authors want to tell in the supplementary section "Determining the 28 transcription factors ..."*

This supplementary section has been moved right after “Criteria and full results for predicting transcriptional regulation under 51 environmental conditions”. It fits better here as this new section directly references “Determining the 28 transcription factors...” section.

3rd Editorial Decision

24 April 2014

Thank you again for submitting your work to Molecular Systems Biology. We have now heard back from the two referees who agreed to evaluate your manuscript. In addition to the previous reviewer #1 (current reviewer #1) who was asked to examine whether their previous comments have been addressed, we invited a new referee (#2), who evaluated the study afresh. As you will see from the reports below, referee #1 thinks that the previous comments have been satisfactorily addressed. However, referee #2 raises a series of concerns, which should be carefully addressed in a revision of the manuscript.

Without repeating all the comments listed below, the more fundamental points are the following:

- More systematic evaluation of the novel predictions on TF regulatory activities is required to better support the predictive value of the MinSpan pathways.
- The comparison of MinSpan pathways to existing pathway databases needs to be extended and documented in better detail.

Moreover, several of the comments of reviewer #2 refer to the need to better document several points throughout the manuscript.

If you feel you can satisfactorily deal with these points and those listed below, you may wish to submit a revised version of your manuscript. Please attach a covering letter providing details of the way in which you have handled each of the points raised by the referees. A revised manuscript will be once again subject to review and you probably understand that we can give you no guarantee at this stage that the eventual outcome will be favorable.

Reviewer #1:

I am satisfied how the authors dealt with my comments to the previous version of the manuscript.

Reviewer #2:

In this manuscript by Bordbar et al., the authors describe a new method to create a parsimonious pathway structure for genome-scale metabolic networks. The authors compare their method with both human-defined and automatically-derived pathways and found it to better describe genetic, protein and regulatory interactions. Finally, they perform growth and transcriptional profile experiments to validate three of their TF predictions, two of which are indeed validated based on their criteria.

The manuscript is well-written and the topic is of high significance. This work builds upon the previous work of the same group on extreme pathways and elementary flux models and extends it significantly in a way that the number of pathways that completely cover the phenotypic space is minimized to thousands instead of billions or more, which was the case in extreme pathways. However, there are major concerns regarding the analysis and a number of points that need to be addressed to assess the contribution of this manuscript. More specifically:

1.a. One of the main claims of this paper is that analysis of co-occurrence of MinSpan pathways can predict with "high accuracy and coverage" protein, genetic and regulatory interactions. Although the main manuscript (140-173, M&M) and Suppl. Text (sections 2 and 3) provide details on how this analysis, I find them to be incomplete and a major effort has to be put to have organized M&M and Suppl. Mat. (which is rather short and not well organized) subsections that explain each of the following points.

1.b. First, it is not clear what was taken as ground truth from the various databases. For example for TF-gene interactions, the authors cite RegulonDB (628), but it is not clear which group set (confirmed?strong? weak-evidence interactions? what is the negative set? Everything that is not reported?).

1.c. Second, how the TF/TN and the AUC are calculated is not clear. Equally not clear is whether the authors constructed an ROC curve and a PR curve with a convex hull (necessary), which have to be reported and added at least as a suppl. figure.

1.d. The comparison of the MinSpan with other DBs is for different # of interactions (Fig. 2D-F) which render it inconclusive at best. The authors should re-do the analysis and report the predictive capacity of MinSpan across multiple testing sets that however are kept the same when comparing the various methods and for the same specificity/accuracy points. In addition, points of how well current methods fare should be added on the 2D-F plots.

2. Statistics do not accompany many of the claims, for example 159-163, there are no p-values or support for "accurately predict", same with 191-199, substantiate all comparisons/arguments throughout the text.

3. A sensitivity analysis has to be performed to evaluate the effect of the model parameters (including the ub/lb, which, for no reason - other than perhaps FBA legacy -were picked to be +/- 1000) on the MinSpan results.

4. The validation is performed in the case of 3 TFs only (3 RNA-Seq samples), with 2 predictions, actually accepted as correct. Obviously this is not adequate and a more extended validation should be performed to provide some more confidence that the method works.

5. Other comments:

5.1. There is some repetition in 140-147 please revise. Same with 150-154.

5.2. In 168, any 30 genes, or only the ones related to metabolism?

5.3. In 212, which databases? Revise 211-216, statements are unclear.

5.4. In 271, Minspan enumerates pathways regardless on whether they are described in a database, correct?

5.5. In 278- 290, please reference Fig. 4B.

5.6. In 387, not clear what "uniquely" refers to. It is not clear from this sentence on how they predict TF activity from a DE gene set. Lines 391-393 provide some hint but again not clear how predictions are reported as correct or indeterminate.

5.7 In 394, the authors report that "the gene set was enriched with GcvA, Lrp and PurR" and they report three p-values. A gene set can be "enriched" with specific biological group(s), but not clear what this means in the case of one protein or what the p-values correspond to, so please revise.

5.8. In M&M, the authors should briefly explain in M&M or main text what the null space is (i.e. set of solutions) for clarity and define b_i .

5.9. In 574, what is the "coverage criteria"? Are the authors referring to the log(interactions) of Fig. 2D-F?

Thank you again for the consideration of our manuscript titled: "Minimal metabolic pathway structure is consistent with associated biomolecular interactions". We have enclosed a revised version of the manuscript, which fully addresses the raised concerns by Reviewer #2.

We found the reviewer's comments to be helpful and constructive in sharpening the content of the manuscript. Revisions were made throughout the manuscript and each point brought up by the reviewers was addressed through revision of the manuscript, when appropriate, and a response (please see below).

Among the revisions, we have strengthened the statistics for the correlation analysis as requested. We have also added a more systematic evaluation of the transcription factor predictions.

I believe that we have addressed all of the concerns raised and we hope that you find the revisions and accompanying explanation satisfactory. If there are any points or issues, which I could offer further clarification on, please do not hesitate to contact me.

We would like to thank Reviewer #1 for their previous constructive comments.

Reviewer #1:

I am satisfied how the authors dealt with my comments to the previous version of the manuscript.

We would like to thank Reviewer #2 on their constructive comments to further clarify the text and methods, as well as further strengthen the statistical evaluations.

Reviewer #2:

In this manuscript by Bordbar et al., the authors describe a new method to create a parsimonious pathway structure for genome-scale metabolic networks. The authors compare their method with both human-defined and automatically-derived pathways and found it to better describe genetic, protein and regulatory interactions. Finally, they perform growth and transcriptional profile experiments to validate three of their TF predictions, two of which are indeed validated based on their criteria.

The manuscript is well-written and the topic is of high significance. This work builds upon the previous work of the same group on extreme pathways and elementary flux models and extends it significantly in a way that the number of pathways that completely cover the phenotypic space is minimized to thousands instead of billions or more, which was the case in extreme pathways. However, there are major concerns regarding the analysis and a number of points that need to be addressed to assess the contribution of this manuscript. More specifically:

1.a. One of the main claims of this paper is that analysis of co-occurrence of MinSpan pathways can predict with "high accuracy and coverage" protein, genetic and regulatory interactions. Although the main manuscript (140-173, M&M) and Suppl. Text (sections 2 and 3) provide details on how this analysis, I find them to be incomplete and a major effort has to be put to have organized M&M and Suppl. Mat. (which is rather short and not well organized) subsections that explain each of the following points.

1.b. First, it is not clear what was taken as ground truth from the various databases. For example for TF-gene interactions, the authors cite RegulonDB (628), but it is not clear which group set (confirmed? strong? weak-evidence interactions? what is the negative set? Everything that is not reported?).

We have updated the "Correlation Analysis" section in M&M to clarify these points. Specifically, we used all reported TF-gene interactions in RegulonDB. The negative set in the case of all comparisons was a lack of reported biomolecular interactions.

1.c. Second, how the TF/TN and the AUC are calculated is not clear. Equally not clear is whether the authors constructed an ROC curve and a PR curve with a convex hull (necessary), which have to be reported and added at least as a suppl. figure.

The ROC curve plots (with a convex hull) have been added to the supplementary material. The PR curves have been added as well. It is important to note that for the analysis presented, the PR curve is not as important as we are not looking to “retrieve” or predict biomolecular interactions, but instead to see if the pathways are representative of interactions. This means that both interactions AND the lack of interactions are important, which the PR curve does not take into account (PR curve does not look at true negatives). In hindsight, the use of the word “predict” is not entirely correct for what the correlation analysis is supposed to test. We have removed “predict” in the correlation analysis section and used instead what most of the text uses which is “representative” of the underlying biomolecular interactions.

1.d. The comparison of the MinSpan with other DBs is for different # of interactions (Fig. 2D-F) which render it inconclusive at best. The authors should re-do the analysis and report the predictive capacity of MinSpan across multiple testing sets that however are kept the same when comparing the various methods and for the same specificity/accuracy points. In addition, points of how well current methods fare should be added on the 2D-F plots.

The different databases have pathways with different metabolic genes and proteins, hence the different number of interactions predicted. Therefore, the only true comparison would be the intersection of the # of interactions. Unfortunately, the intersection is so small (89 total tests – PPI, 102 total tests – positive genetic interactions, 51 total tests – TRN local, 137 total tests – TRN int, 152 total tests – TRN global) that the analysis would not be characteristic compared to all of metabolism (Fig 2D-F have total tests for MinSpan and GO in 10^4).

However to address this comment, we have repeated all correlation analyses, including ROC and PR curves, for the particular interactions of each databases coverage, as well as the union of all the interactions. When a particular database does not have coverage of that particular biomolecular interaction, a correlation of 0 was assumed. This artificial scoring skews the results for each database but is the only reasonable way to redo the analysis over the same numbers of interactions. Still, MinSpan is overall as representative or more representative of the underlying biomolecular interactions than the three other databases. These results have been added to a supplementary section and figures. These results have not been added to Figure 2D-F, as an additional 13 points (and captions) are required on each of those panels, which would make the figure nearly undecipherable.

2. Statistics do not accompany many of the claims, for example 159-163, there are no p-values or support for "accurately predict", same with 191-199, substantiate all comparisons/arguments throughout the text.

p-values have been added in comparison of the ROC curves in the main text as well as for the ROC curves in the supplemental sections. As previously qualitatively stated in the text, there was no significance for the protein interactions, but there were statistical significance in differences in the ROC curves for the genetic interactions and transcriptional regulation. We have also added these same p-values for all of the interactions tested with different numbers of interactions (as discussed in 1d above).

3. A sensitivity analysis has to be performed to evaluate the effect of the model parameters (including the ub/lb, which, for no reason - other than perhaps FBA legacy -were picked to be +/- 1000) on the MinSpan results.

The calculation of the MinSpan pathways is dependent only on the directionality of intracellular metabolic reactions and transporters, which are based on thermodynamics and enzyme properties. The algorithm is run on a network with all metabolic exchanges open to encapsulate the entire solution space during calculations. The magnitude of the lb/ub (if they are 1000, 10, 0.1, etc.) does not affect the MinSpan calculation and hence a sensitivity analysis is not required.

However, this does bring up an important point about how the model structure is important for the MinSpan calculations. This is already discussed in the manuscript in the limitation portion of the Discussion section.

4. The validation is performed in the case of 3 TFs only (3 RNA-Seq samples), with 2 predictions, actually accepted as correct. Obviously this is not adequate and a more extended validation should be performed to provide some more confidence that the method works.

We apologize for not being fully clear on the validation of TF-environment associations. Across the 51 environmental shifts, there were 283 TF-environmental shift associations predicted. The enrichment with known TF associations in the literature is highly significant ($p = 2.79e-107$, binomial distribution). This was not explicitly stated in the manuscript previously. Looking specifically by shifts, 37 matched known associations and 8 partially matched known associations. For discovering new regulation, we focused on these 45 shifts. There are a total of 247 predictions for the 45 shifts. We did not explicitly state in the previous text, but for these 45 shifts, 153 of the predictions are confirmed in EcoCyc and primary literature. This is already quite a large validation of the approach. Text explicitly stating these literature validations has been added at Lines 381-388.

We then decided to look at the 93 unconfirmed regulatory associations to determine if some of them are in fact undiscovered associations rather than false positives. As is already in the text, we have experimentally tested 3 of the 93 predictions. Further, we have already started using the MinSpan predictions to further study the *E. coli* TRN. A manuscript is in preparation where we used two additional MinSpan predictions (i.e. Nac/cytosine and GlnG/cytosine) to study regulation of nitrogen metabolism. The environmental shift induces the transcription factor, as determined by Western blot. In addition, ChIP with exome sequencing was done to determine novel regulatory binding sites. Binding sites related to the prediction were confirmed for Nac and NtrC. Further, the confirmed binding sites overlapped with some of the ones inferred by RNA-Seq in Figure 6 for Nac. The manuscript in preparation is part of Donghyuk Kim's PhD thesis, which has been printed and the thesis has been cited in the main text. This is an additional, and much more in-depth, experimental validation.

Finally, addition of new RNA-Seq experiments is outside of the scope of this study. Every additional TF-environmental association validation is not 1 RNA-Seq experiment as the Reviewer suggests, but 3 additional RNA-Seq experiments (4 different conditions, but the WT results can be reused). Doing a power analysis, if we assume that the predictive rate of the method remains at 66%, one would need to do 27 experiments in total to be statistically more significant than a 50% predictive rate. This would entail an additional $22 \times 3 = 66$ RNA-Seq experiments. Each RNA-Seq run costs us roughly \$1100 for MiSeq reagents, excluding all preparatory and personnel cost. Further, additional experiments could potentially warrant multiple publications depending on the number of new regulatory interactions discovered, as evidenced by the manuscript above. Thus, we believe new experiments are outside of the scope of this study.

5. Other comments:

5.1. There is some repetition in 140-147 please revise. Same with 150-154.

The text has been updated in both locations to remove repetition. Specifically, we reordered 140-147 to remove repetition. For 150-154, we removed the first sentence.

5.2. In 168, any 30 genes, or only the ones related to metabolism?

Yes, these are metabolism related genes. The text has been updated to clarify this.

5.3. In 212, which databases? Revise 211-216, statements are unclear.

We have updated the second sentence of that paragraph to mention the specific databases so the rest of the paragraph reads better.

5.4. In 271, Minspan enumerates pathways regardless on whether they are described in a database, correct?

Yes, that is what is written at Line 271. We have not made any changes for this sub-point as we believe the text is already clear in this section.

5.5. *In 278- 290, please reference Fig. 4B.*

Figure 4B was already referenced in the second sentence of this particular section. We have moved the reference to the next sentence, as it fits better there.

5.6. *In 387, not clear what "uniquely" refers to. It is not clear from this sentence on how they predict TF activity from a DE gene set. Lines 391-393 provide some hint but again not clear how predictions are reported as correct or indeterminate.*

The sentence has been reworded for clarity. Further, that sentence contains a reference to the Materials and Methods. In particular, the section titled "Analysis workflow for dual perturbations" explicitly defines the particular gene set.

5.7 *In 394, the authors report that "the gene set was enriched with GcvA, Lrp and PurR" and they report three p-values. A gene set can be "enriched" with specific biological group(s), but not clear what this means in the case of one protein or what the p-values correspond to, so please revise.*

The sentence has been reworded for clarity to: "the gene set was enriched with genes known to be regulated by the TFs GcvA, Lrp, and PurR".

The three proteins are transcription factors and the differential gene set was enriched in genes that are known to be regulated by those particular transcription factors.

5.8. *In M&M, the authors should briefly explain in M&M or main text what the null space is (i.e. set of solutions) for clarity and define b_i .*

The definition of the null space is already described in the Introduction (Lines 73-77). We have changed one of the sentences to add further clarity for this sub-point.

"Thus, all potential steady-state reaction fluxes of a metabolic network are contained in the associated null space of S ."

Has been changed to:

"Thus, the full set of potential steady-state reaction fluxes of a metabolic network is contained..."

Further, we have added a sentence after the MinSpan pseudo-code defining the b vector.

5.9. *In 574, what is the "coverage criteria"? Are the authors referring to the log(interactions) of Fig. 2D-F?*

Yes, Line 574 in the M&M is explicitly describing what the "log(interactions)" is for Figure 2. We have clarified this point in the text.

Thank you again for sending us your revised manuscript. As you will see from the comments below, Reviewer #2 is now satisfied with the modifications made and I am pleased to inform you that your paper has been accepted for publication.

Thank you very much for submitting your work to Molecular Systems Biology.

Reviewer #2:

The authors have addressed adequately my concerns and have revised the manuscript to include the necessary information.

On another note, I still think that a small-scale experimental validation is in order. The authors' analysis regarding 66 RNA-Seq lanes that would cost \$72K is wrong - actually one multiplexed RNA-Seq SR50 HiSeq2500 lane for 30-35 samples will give adequate coverage for validating 10 predictions (3 conditions + shared WT) and it would cost \$1500, with reagents and adapters. Since this will delay the publication for about 1.5 months and it is not absolutely necessary, I am fine with publishing it as is at the editor's discretion.